# DATA AUGMENTATION AS STOCHASTIC OPTIMIZATION

## ABSTRACT

We present a theoretical framework recasting data augmentation as stochastic optimization for a sequence of time-varying proxy losses. This provides a unified language for understanding techniques commonly thought of as data augmentation, including synthetic noise and label-preserving transformations, as well as more traditional ideas in stochastic optimization such as learning rate and batch size scheduling. We then specialize our framework to study arbitrary augmentations in the context of a simple model (overparameterized linear regression). We extend in this setting the classical Monro-Robbins theorem to include augmentation and obtain rates of convergence, giving conditions on the learning rate and augmentation schedule under which augmented gradient descent converges. Special cases give provably good schedules for augmentation with additive noise, minibatch SGD, and minibatch SGD with noise.

## 1  INTRODUCTION

Implementing gradient-based optimization in practice requires many choices. These include setting hyperparameters such as learning rate and batch size as well as specifying a data augmentation scheme, a popular set of techniques in which data is augmented (i.e. modified) at every step of optimization. Trained model quality is highly sensitive to these choices. In practice they are made using methods ranging from a simple grid search to Bayesian optimization and reinforcement learning (Cubuk et al., 2019; 2020; Ho et al., 2019). Such approaches, while effective, are often ad-hoc and computationally expensive due to the need to handle scheduling, in which optimization hyperparameters and augmentation choices and strengths are chosen to change over the course of optimization.

These empirical results stand in contrast to theoretically grounded approaches to stochastic optimization which provide both provable guarantees and reliable intuitions. The most extensive work in this direction builds on the seminal article (Robbins & Monro, 1951), which gives provably optimal learning rate schedules for stochastic optimization of strongly convex objectives. While rigorous, these approaches are typically are not sufficiently flexible to address the myriad augmentation types and hyperparameter choices beyond learning rates necessary in practice.

This article is a step towards bridging this gap. We provide in §3 a rigorous framework for reinterpreting gradient descent with arbitrary data augmentation as stochastic gradient descent on a time-varying sequence of objectives. This provides a unified language to study traditional stochastic optimization methods such as minibatch SGD together with widely used augmentations such as additive noise (Grandvalet & Canu, 1997), CutOut (DeVries & Taylor, 2017), Mixup (Zhang et al., 2017) and label-preserving transformations (e.g. color jitter, geometric transformations (Simard et al., 2003)). It also opens the door to studying how to schedule and evaluate *arbitrary* augmentations, an important topic given the recent interest in learned augmentation Cubuk et al. (2019).

Quantitative results in our framework are difficult to obtain in full generality due to the complex interaction between models and augmentations. To illustrate the utility of our approach and better understand specific augmentations, we present in §3 and §5 results about arbitrary augmentations for overparameterized linear regression and specialize to additive noise and minibatch SGD in §4 and §6. While our results apply directly only to simple quadratic losses, they treat very general augmentations. Treating more complex models is left to future work. Our main contributions are:

- In Theorem 5.1, we give sufficient conditions under which gradient descent under *any* augmentation scheme converges in the setting of overparameterized linear regression. Our

result extends classical results of Monro-Robbins type and covers schedules for both learning rate and data augmentation scheme.

- We complement the asymptotic results of Theorem 5.1 with quantitative rates of convergence furnished in Theorem 5.2. These rates depend only on the first few moments of the augmented data distribution, underscoring the flexibility of our framework.

- In §4, we analyze additive input noise, a popular augmentation strategy for increasing model robustness. We recover the known fact that it is equivalent to stochastic optimization with $\ell_2$-regularization and find criteria in Theorem 4.1 for jointly scheduling the learning rate and noise level to provably recover the minimal norm solution.

- In §6, we analyze minibatch SGD, recovering known results about rates of convergence for SGD (Theorem 6.1) and novel results about SGD with noise (Theorem 6.2).

## 2 RELATED WORK

In addition to the extensive empirical work on data augmentation cited elsewhere in this article, we briefly catalog other theoretical work on data augmentation and learning rate schedules. The latter were first considered in the seminal work Robbins & Monro (1951). This spawned a vast literature on *rates* of convergence for GD, SGD, and their variants. We mention only the relatively recent articles Bach & Moulines (2013); Défossez & Bach (2015); Bottou et al. (2018); Smith et al. (2018); Ma et al. (2018) and the references therein. The last of these, namely Ma et al. (2018), finds optimal choices of learning rate and batch size for SGD in the overparametrized linear setting.

A number of articles have also pointed out in various regimes that data augmentation and more general transformations such as feature dropout correspond in part to $\ell_2$-type regularization on model parameters, features, gradients, and Hessians. The first article of this kind of which we are aware is Bishop (1995), which treats the case of additive Gaussian noise (see §4). More recent work in this direction includes Chapelle et al. (2001); Wager et al. (2013); LeJeune et al. (2019); Liu et al. (2020). There are also several articles investigating *optimal* choices of $\ell_2$-regularization for linear models (cf e.g. Wu et al. (2018); Wu & Xu (2020); Bartlett et al. (2020)). These articles focus directly on the generalization effects of ridge-regularized minima but not on the dynamics of optimization. We also point the reader to Lewkowycz & Gur-Ari (2020), which considers optimal choices for the weight decay coefficient empirically in neural networks and analytically in simple models.

We also refer the reader to a number of recent attempts to characterize the benefits of data augmentation. In Rajput et al. (2019), for example, the authors quantify how much augmented data, produced via additive noise, is needed to learn positive margin classifiers. Chen et al. (2019), in contrast, focuses on the case of data invariant under the action of a group. Using the group action to generate label-preserving augmentations, the authors prove that the variance of any function depending only on the trained model will decrease. This applies in particular to estimators for the trainable parameters themselves. Dao et al. (2019) shows augmented $k$-NN classification reduces to a kernel method for augmentations transforming each datapoint to a finite orbit of possibilities. It also gives a second order expansion for the proxy loss of a kernel method under such augmentations and interprets how each term affects generalization. Finally, the article Wu et al. (2020) considers both label preserving and noising augmentations, pointing out the conceptually distinct roles such augmentations play.

## 3 DATA AUGMENTATION AS STOCHASTIC OPTIMIZATION

A common task in modern machine learning is the optimization of an empirical risk

$$\mathcal{L}(W; \mathcal{D}) = \frac{1}{|\mathcal{D}|} \sum_{(x_j, y_j) \in \mathcal{D}} \ell(f(x_j; W), y_j), \tag{3.1}$$

where $f(x; W)$ is a parameterized model for a dataset $\mathcal{D}$ of input-response pairs $(x, y)$ and $\ell$ is a per-sample loss. Optimizing $W$ by vanilla gradient descent on $\mathcal{L}$ corresponds to the update equation

$$W_{t+1} = W_t - \eta_t \nabla_W \mathcal{L}(W_t; \mathcal{D}).$$

In this context, we define a *data augmentation scheme* to be any procedure that consists, at every step of optimization, of replacing the dataset $\mathcal{D}$ by a randomly augmented variant, which we will denote

by $\mathcal{D}_t$. Typically, $\mathcal{D}_t$ is related to $\mathcal{D}$ in some way, but our framework does not explicitly constrain the form of this relationship. Instead, certain conditions on this relationship will be required for our main results Theorems 5.1 and 5.2 to give useful results for a specific augmentation scheme. A data augmentation scheme therefore corresponds to the augmented update equation

$$W_{t+1} = W_t - \eta_t \nabla_W \mathcal{L}(W_t; \mathcal{D}_t). \tag{3.2}$$

Since $\mathcal{D}_t$ is a stochastic function of $\mathcal{D}$, it is natural to view the augmented update rule (3.2) as a form of stochastic optimization for the *proxy loss at time* $t$

$$\overline{\mathcal{L}}_t(W) := \mathbb{E}_{\mathcal{D}_t} \left[ \mathcal{L}(W; \mathcal{D}_t) \right]. \tag{3.3}$$

The update (3.2) corresponds precisely to stochastic optimization for the time-varying objective $\overline{\mathcal{L}}_t(W)$ in which the unbiased estimate of its gradient is obtained by evaluating the gradient of $\mathcal{L}(W; \mathcal{D}_t)$ on a single sample $\mathcal{D}_t$ drawn from the augmentation distribution. The connection between data augmentation and this proxy loss was introduced for Gaussian noise in Bishop (1995) and in general in Chapelle et al. (2001), but we now consider it in the context of stochastic optimization.

Despite being mathematically straightforward, reformulating data augmentation as stochastic optimization provides a unified language for questions about learning rate schedules and general augmentation schemes including SGD. In general, such questions can be challenging to answer, and even evaluating the proxy loss $\overline{\mathcal{L}}_t(W)$ may require significant ingenuity.

While we will return to more sophisticated models in future work, we henceforth analyze general augmentations in the simple context of overparameterized linear regression. Though there are many ways to perform linear regression, we restrict to augmented gradient descent both to gain intuition about specific augmentations and to understand the effect of augmentation on optimization. We therefore consider optimizing the entries of a weight matrix $W \in \mathbb{R}^{p \times n}$ by gradient descent on

$$\mathcal{L}(W; \mathcal{D}) = \frac{1}{|\mathcal{D}|} \sum_{(x,y) \in \mathcal{D}} ||y - Wx||_F^2 = \frac{1}{N} ||Y - WX||_F^2, \tag{3.4}$$

where our dataset $\mathcal{D}$ is summarized by data matrices $X \in \mathbb{R}^{n \times N}$ and $Y \in \mathbb{R}^{p \times N}$, whose $N < n$ columns consist of inputs $x_i \in \mathbb{R}^n$ and associated labels $y_i \in \mathbb{R}^p$. Following this notation, a data augmentation scheme is specified by prescribing at each time step an augmented dataset $\mathcal{D}_t$ consisting of modified data matrices $X_t, Y_t$, whose columns we denote by $x_{i,t} \in \mathbb{R}^n$ and $y_{i,t} \in \mathbb{R}^p$. Here, the number of columns in $X_t$ and $Y_t$ (i.e. the number of datapoints in $\mathcal{D}_t$) may vary.

We now give examples of some commonly used augmentations our framework can address.

- **Additive Gaussian noise:** This is implemented by setting $X_t = X + \sigma_t \cdot G$ and $Y_t = Y$ for $\sigma_t > 0$ and $G$ a matrix of i.i.d. standard Gaussians. We analyze this in §4.

- **Mini-batch SGD:** To implement mini-batch SGD with batch size $B_t$, we can take $X_t = XA_t$ and $Y_t = YA_t$ where $A_t \in \mathbb{R}^{N \times B_t}$ has i.i.d. columns containing a single non-zero entry equal to 1 chosen uniformly at random. We analyze this in detail in §6.

- **Random projection:** This is implemented by $X_t = \Pi_t X$ and $Y_t = Y$, where $\Pi_t$ is an orthogonal projection onto a random subspace. For $\gamma_t = \mathrm{Tr}(\Pi_t)/n$, the proxy loss is

$$\overline{\mathcal{L}}_t(W) = \|Y - \gamma_t WX\|_F^2 + \gamma_t(1 - \gamma_t)n^{-1}\mathrm{Tr}(XX^\mathsf{T})\|W\|_F^2 + O(n^{-1}),$$

  which adds a data-dependent $\ell_2$ penalty and applies a Stein shrinkage on input data.

- **Label-preserving transformations:** For a 2-D image viewed as a vector $x \in \mathbb{R}^n$, geometric transforms (with pixel interpolation) or other label-preserving transforms such as color jitter take the form of linear transforms $\mathbb{R}^n \to \mathbb{R}^n$. We may implement such augmentations in our framework by $X_t = A_t X$ and $Y_t = Y$ for some random transform matrix $A_t$.

- **Mixup:** To implement Mixup, we can take $X_t = XA_t$ and $Y_t = YA_t$, where $A_t \in \mathbb{R}^{N \times B_t}$ has i.i.d. columns containing with two random non-zero entries equal to $1 - c_t$ and $c_t$ with mixing coefficient $c_t$ drawn from a $\mathrm{Beta}(\alpha_t, \alpha_t)$ distribution for a parameter $\alpha_t$.

Our main technical results, Theorems 5.1 and 5.2, give sufficient conditions for a learning rate schedule $\eta_t$ and a schedule for the statistics of $X_t, Y_t$ under which optimization with augmented gradient descent will provably converge. We state these general results in §5. Before doing so, we seek to demonstrate both the utility of our framework and the flavor of our results by focusing on the simple but already informative case of additive Gaussian noise.

## 4 AUGMENTATION WITH ADDITIVE GAUSSIAN NOISE

An common augmentation in practice injects input noise as a regularizer (Grandvalet & Canu, 1997):

$$\mathcal{D}_t = \{(x_{i,t}, y_{i,t}), \ i = 1, \ldots, N\}, \qquad x_{i,t} = x_i + \sigma_t g_{i,t}, \quad y_{i,t} = y_i,$$

where $g_{i,t}$ are i.i.d. standard Gaussian vectors and $\sigma_t$ is a strength parameter. This section studies such augmentations using our framework. A direct computation reveals that the proxy loss

$$\overline{\mathcal{L}}_t(W) = \mathcal{L}_{\sigma_t}(W) := \mathcal{L}(W; \mathcal{D}) + \sigma_t^2 \, ||W||_F^2$$

corresponding to additive Gaussian noise adds an $\ell_2$-penalty to the original loss $\mathcal{L}$. This is simple but useful intuition. It also raises the question: what is the optimal relation between the learning rate $\eta_t$ and the augmentation strength $\sigma_t$ (i.e. the $\ell_2$-penalty)?

To get a sense of what optimal might mean in this context, observe first that if $\sigma_t = 0$, then directly differentiating the loss $\mathcal{L}$ yields the following update rule:

$$W_{t+1} = W_t + \frac{2\eta_t}{N} \cdot (Y - W_t X) X^\mathsf{T}. \tag{4.1}$$

The increment $W_{t+1} - W_t$ is therefore contained in the column span

$$V_\| := \text{column span of } X X^\mathsf{T} \subseteq \mathbb{R}^n \tag{4.2}$$

of the model Hessian $X X^\mathsf{T}$. Overparameterization implies $V_\| \neq \mathbb{R}^n$. The component $W_{t,\perp}$ of $W_t$ that is in the orthogonal complement of $V_\|$ thus remains frozen to its initialized value. Geometrically, this means that there are some directions, namely those in the orthogonal complement to $V_\|$, which gradient descent "cannot see." Optimization with appropriate step sizes then yields

$$\lim_{t \to \infty} W_t = W_{0,\perp} + W_{\min}, \qquad W_{\min} := Y X^\mathsf{T} (X X^\mathsf{T})^+,$$

where $W_{\min}$ is the minimum norm solution of $Y = WX$. The original motivation for introducing the $\ell_2$-regularized losses $\mathcal{L}_\sigma$ is that they provide a mechanism to eliminate the component $W_{0,\perp}$ for all initializations, not just the special choice $W_0 = 0$, and they can be used to regularize non-linear models as well. Indeed, for $\sigma > 0$, the loss $\mathcal{L}_\sigma$ is strictly convex and has a unique minimum

$$W_\sigma^* := Y X^\mathsf{T} \left( X X^\mathsf{T} + \sigma^2 N \cdot \mathrm{Id}_{n \times n} \right)^{-1},$$

which tends to the minimal norm solution in the weak regularization limit $\lim_{\sigma \to 0} W_\sigma^* = W_{\min}$. Geometrically, this is reflected in the fact that $\ell_2$-penalty yields non-trivial gradient updates

$$W_{t+1,\perp} = W_{t,\perp} - \eta_t \sigma^2 W_{t,\perp} = (\mathrm{Id} - \eta_t \sigma^2) W_{t,\perp} = \prod_{s=1}^{t} (\mathrm{Id} - \eta_s \sigma^2) W_{0,\perp}, \tag{4.3}$$

which drive this perpendicular component of $W_t$ to zero provided $\sum_{t=1}^{\infty} \eta_t = \infty$. However, for each positive value of $\sigma$, the $\ell_2$-penalty also modifies the gradient descent updates for $W_{t,\|}$, ultimately causing $W_t$ to converge to $W_\sigma^*$, which is not a minimizer of the original loss $\mathcal{L}$.

This downside of ridge regression motivates jointly scheduling the step size $\eta_t$ and the noise strength $\sigma_t$. We hope that driving $\sigma_t$ to 0 at an appropriate rate can guarantee convergence of $W_t$ to $W_{\min}$. Namely, we want to retain the regularizing effects of $\ell_2$-noise to force $W_{t,\perp}$ to zero while mitigating its adverse effects which prevent $W_\sigma^*$ from minimizing $\mathcal{L}$. It turns out that this is indeed possible:

**Theorem 4.1** (Special case of Theorem 5.1). *Suppose $\sigma_t^2, \eta_t \to 0$ with $\sigma_t^2$ non-increasing and*

$$\sum_{t=0}^{\infty} \eta_t \sigma_t^2 = \infty \qquad and \qquad \sum_{t=0}^{\infty} \eta_t^2 \sigma_t^2 < \infty. \tag{4.4}$$

*Then, $W_t \xrightarrow{p} W_{min}$. Further, if $\eta_t = \Theta(t^{-x})$ and $\sigma_t^2 = \Theta(t^{-y})$ with $x, y > 0$, $x + y < 1$, and $2x + y > 1$, then for any $\epsilon \in (0, \min\{y, x/2\})$, we have that*

$$t^{\min\{y, \frac{1}{2}x\} - \epsilon} ||W_t - W_{min}||_F \xrightarrow{p} 0.$$

Let us give a few comments on Theorem 4.1. First, although it is stated for additive Gaussian noise, an analogous version holds for arbitrary additive noise with bounded moments, with the only change being a constant multiplicative factor in the second condition of (4.4).

Second, that convergence in probability $W_t \xrightarrow{p} W_{\min}$ follows from (4.4) is analogous to a Monro-Robbins type theorem (Robbins & Monro, 1951). Indeed, inspecting (4.3), we see that the first condition in (4.4) guarantees that the effective learning rate $\eta_t \sigma_t^2$ in the orthogonal complement to $V_\parallel$ is sufficiently large that the corresponding component $W_{t,\perp}$ of $W_t$ tends to 0, allowing the result of optimization to be independent of the initial condition $W_0$. Further, the second condition in (4.4) guarantees that the variance of the gradients, which at time $t$ scales like $\eta_t^2 \sigma_t^2$ is summable. As in the usual Monro-Robbins setup, this means that only a finite amount of noise is injected into the optimization. Further, (4.4) is a direct specialization of (5.5) and (5.6) from Theorem 5.1.

Third, by optimizing over $x, y$, we see that fastest rate of convergence guaranteed by Theorem 4.1 is obtained by setting $\eta_t = t^{-2/3+\epsilon}$, $\sigma_t^2 = t^{-1/3}$ and results in a $O(t^{-1/3+\epsilon})$ rate of convergence. It is not evident that this is the best possible rate, however.

Finally, although we leave systemic study of augmentation in non-linear models to future work, our framework can be applied beyond linear models and quadratic losses. To see this, as noted for kernels in Dao et al. (2019), augmenting inputs of nonlinear feature models correspond to applying different augmentations on the outputs of the feature map. To give a concrete example, consider additive noise for small $\sigma_t$. For any sufficiently smooth function $g$, Taylor expansion reveals

$$\mathbb{E}\left[g(x + \sigma_t G)\right] = g(x) + \frac{\sigma_t^2}{2}\Delta g(x) + O(\sigma_t^4),$$

where $\Delta = \sum_i \partial_i^2$ is the Laplacian and $G$ is a standard Gaussian vector. For a general loss of the form (3.1) we have

$$\overline{\mathcal{L}}_t(W) = \mathcal{L}(W; \mathcal{D}) + \frac{\sigma_t^2}{2|\mathcal{D}|} \sum_{(x,y \in \mathcal{D})} \mathrm{Tr}\left[(\nabla_x f)^\mathsf{T} (H_f \ell) \nabla_x f\right] + (\nabla_f \ell)^\mathsf{T} \Delta_x f + O(\sigma_t^4),$$

where we have written $H_f \ell$ for the Hessian of some convex per-sample loss $\ell$ with respect to $f$ and $\nabla_x, \nabla_f$ for the gradients with respect to $x, f$, respectively. This is consistent with the similar expansion done in the kernel setting by Dao et al. (2019, Section 4). If $\sigma_t$ is small, then the proxy loss $\overline{\mathcal{L}}_t$ will differ significantly from the unaugmented loss $\mathcal{L}$ only near the end of training, when we expect $\nabla_f \ell$ to be small and $H_f \ell$ to be positive semi-definite. Hence, we find heuristically that, neglecting higher order terms in $\sigma_t$, additive noise with small $\sigma_t$ corresponds to an $\ell_2$-regularizer

$$\mathrm{Tr}\left[\frac{\sigma_t^2}{2}(\nabla_x f)^\mathsf{T}(H_f \mathcal{L})\nabla_x f\right] =: \frac{\sigma_t^2}{2}\|\nabla_x f\|_{H_f \mathcal{L}}^2$$

for the gradients of $f$ with respect to the natural inner product determined by the Hessian of the loss. This is intuitive since penalizing the gradients of $f$ is the same as requiring that $f$ is approximately constant in a neighborhood of every datapoint. However, although the input noise was originally isotropic, the $\ell_2$-penalty is aligned with the loss Hessian and hence need not be.

## 5 TIME-VARYING MONRO-ROBBINS FOR LINEAR MODELS UNDER AUGMENTATION

In this section, we state two general results, Theorems 5.1 and 5.2, which provide sufficient conditions for jointly scheduling learning rates and general augmentation schemes to guarantee convergence of augmented gradient descent in the overparameterized linear model (3.4).

### 5.1 A GENERAL TIME-VARYING MONRO-ROBBINS THEOREM

Given an augmentation scheme for the model (3.4), the time $t$ gradient update at learning rate $\eta_t$ is

$$W_{t+1} := W_t + \frac{2\eta_t}{N} \cdot (Y_t - W_t X_t) X_t^\mathsf{T}, \tag{5.1}$$

where $\mathcal{D}_t = (X_t, Y_t)$ is the augmented dataset at time $t$. The minimum norm minimizer of the corresponding proxy loss $\overline{\mathcal{L}}_t$ (see 3.3) is

$$W_t^* := \mathbb{E}[Y_t X_t^\mathsf{T}] \mathbb{E}[X_t X_t^\mathsf{T}]^+, \tag{5.2}$$

where $\mathbb{E}[X_t X_t^\mathsf{T}]^+$ denotes the Moore-Penrose pseudo-inverse. In this section we state a rigorous result, Theorem 5.1, giving sufficient conditions on the learning rate $\eta_t$ and distributions of the augmented matrices $X_t, Y_t$ under which augmented gradient descent converges. In analogy with the case of Gaussian noise, (5.1) shows $W_{t+1} - W_t$ is contained in the column span of the Hessian $X_t X_t^\mathsf{T}$ of the augmented loss and almost surely belongs to the subspace

$$V_\| := \text{column span of } \mathbb{E}[X_t X_t^\mathsf{T}] \subseteq \mathbb{R}^n. \tag{5.3}$$

To ease notation, we assume that $V_\|$ is independent of $t$. This assumption is valid for additive Gaussian noise, random projection, MixUp, SGD, and their combinations. We explain in Remark B.2 how to generalize Theorems 5.1 and 5.2 to the case where $V_\|$ varies with $t$.

Let us denote by $Q_\| : \mathbb{R}^n \to \mathbb{R}^n$ the orthogonal projection onto $V_\|$. At time $t$, gradient descent leaves the projection $W_t(\mathrm{Id} - Q_\|)$ of $W_t$ onto the orthogonal complement of $V_\|$ unchanged. In contrast, $||W_t Q_\| - W_t^*||_F$ decreases at a rate governed by the smallest positive eigenvalue

$$\lambda_{\min, V_\|}\left(\mathbb{E}\left[X_t X_t^\mathsf{T}\right]\right) := \lambda_{\min}\left(Q_\| \mathbb{E}\left[X_t X_t^\mathsf{T}\right] Q_\|\right)$$

of the Hessian for the proxy loss $\overline{\mathcal{L}}_t$, which is obtained by restricting its full Hessian $\mathbb{E}\left[X_t X_t^\mathsf{T}\right]$ to $V_\|$. Moreover, whether and at what rate $W_t Q_\| - W_t^*$ converges to $0$ must depend on how quickly

$$\Xi_t^* := W_{t+1}^* - W_t^* \tag{5.4}$$

tends to zero. Indeed, $||\Xi_t^*||_F$ is the distance between proxy loss optima at different times and hence must tend to zero if $||W_t Q_\| - W_t^*||_F$ converges to zero.

**Theorem 5.1.** *Suppose that $V_\|$ is independent of $t$, that the learning rate satisfies $\eta_t \to 0$, that the proxy optima satisfy*

$$\sum_{t=0}^\infty ||\Xi_t^*||_F < \infty, \tag{5.5}$$

*ensuring the existence of a limit $W_\infty^* := \lim_{t \to \infty} W_t^*$, and that*

$$\sum_{t=0}^\infty \eta_t \lambda_{min, V_\|}(\mathbb{E}[X_t X_t^\mathsf{T}]) = \infty. \tag{5.6}$$

*If either*

$$\sum_{t=0}^\infty \eta_t^2 \mathbb{E}\left[||X_t X_t^\mathsf{T} - \mathbb{E}[X_t X_t^\mathsf{T}]||_F^2 + ||Y_t X_t^\mathsf{T} - \mathbb{E}[Y_t X_t^\mathsf{T}]||_F^2\right] < \infty \tag{5.7}$$

*or the more refined condition*

$$\sum_{t=0}^\infty \eta_t^2 \mathbb{E}\left[||X_t X_t^\mathsf{T} - \mathbb{E}[X_t X_t^\mathsf{T}]||_F^2 + ||\mathbb{E}[W_t](X_t X_t^\mathsf{T} - \mathbb{E}[X_t X_t^\mathsf{T}]) - (Y_t X_t^\mathsf{T} - \mathbb{E}[Y_t X_t^\mathsf{T}])||_F^2\right] < \infty \tag{5.8}$$

*hold, then for any initialization $W_0$ we have $W_t Q_\| \xrightarrow{p} W_\infty^*$.*

The conditions of Theorem 5.1 can be applied to the choice of joint schedule for the learning rate and augmentation scheme applied to gradient descent. If the same augmentation is applied with different strength parameters at each step $t$ such as $\sigma_t$ for Gaussian noise, they impose conditions on the choice of joint schedule for $\eta_t$ and these strength parameters. In the example of Theorem 4.1 for Gaussian noise, the condition that $\sigma_t^2$ is non-increasing implies (5.5), the first condition of (4.4) implies (5.6), and the second condition of (4.4) implies (5.7).

In addition to the conditions Theorem 5.1 imposes on $\mathcal{D}_t$, the proxy optima $W_t^*$ and their limit $W_\infty^*$ are determined by the distribution of $\mathcal{D}_t$. Therefore, for $W_\infty^*$ in Theorem 5.1 to be a desirable set of parameters for the original dataset $\mathcal{D}$, the augmented dataset $\mathcal{D}_t$ must have some relation to $\mathcal{D}$.

When the augmentation procedure is static in $t$, Theorem 5.1 reduces to a standard Monro-Robbins theorem Robbins & Monro (1951) for the (static) proxy loss $\overline{\mathcal{L}}_t(W)$. As in that setting, condition (5.6) enforces that the learning trajectory travels far enough to reach an optimum. Condition (5.7) implies the weaker condition (5.8); the second summand in (5.8) is the variance of the gradient of the augmented loss $\mathcal{L}(W; \mathcal{D}_t)$, meaning (5.8) implies the total variance of the stochastic gradients is summable. Condition (5.5) is new; it enforces that the minimizers $W_t^*$ of the proxy losses $\overline{\mathcal{L}}_t(W)$ change slowly enough that the augmented optimization procedure can keep pace.

Though it may be surprising that $\mathbb{E}[W_t]$ appears in this condition, it may be interpreted as the gradient descent trajectory for the deterministic sequence of proxy losses $\overline{\mathcal{L}}_t(W)$. Accounting for the dependence on $\mathbb{E}[W_t]$ allows us to give more precise rates using the variance of the stochastic gradient in (5.8); we include both (5.7) and (5.8) to allow a user of our results to separately analyze $\mathbb{E}[W_t]$ to obtain stronger conclusions.

## 5.2 Convergence rates and scheduling for data augmentation

A more precise analysis of the the proof of Theorem 5.1 allows us to obtain rates of convergence for the projections $W_t Q_\parallel$ of the weights onto $V_\parallel$ to the limiting optimum $W_\infty^*$. In particular, when the quantities in Theorem 5.1 have power law decay, we obtain the following result.

**Theorem 5.2** (informal - Special case of Theorem B.4). *If $V_\parallel$ is independent of $t$, the learning rate satisfies $\eta_t \to 0$, and for some $0 < \alpha < 1 < \beta_1, \beta_2$ and $\gamma > \alpha$ we have*

$$\eta_t \lambda_{min, V_\parallel}(\mathbb{E}[X_t X_t^\mathsf{T}]) = \Omega(t^{-\alpha}), \qquad \|\Xi_t^*\|_F = O(t^{-\beta_1}) \tag{5.9}$$

*and*

$$\eta_t^2 \mathbb{E}[\|X_t X_t^\mathsf{T} - \mathbb{E}[X_t X_t^\mathsf{T}]\|_2^2] = O(t^{-\gamma}) \tag{5.10}$$

*and*

$$\eta_t^2 \mathbb{E}\left[\|\mathbb{E}[W_t](X_t X_t^\mathsf{T} - \mathbb{E}[X_t X_t^\mathsf{T}]) - (Y_t X_t^\mathsf{T} - \mathbb{E}[Y_t X_t^\mathsf{T}])\|_F^2\right] = O(t^{-\beta_2}), \tag{5.11}$$

*then for any initialization $W_0$, we have for any $\epsilon > 0$ that*

$$t^{\min\{\beta_1 - 1, \frac{\beta_2 - \alpha}{2}\} - \epsilon} \|W_t Q_\parallel - W_\infty^*\|_F \xrightarrow{p} 0.$$

Theorem 5.2 measures rates in terms of optimization steps $t$, but a different measurement of time called the *intrinsic time* of the optimization will be more suitable for measuring the behavior of optimization quantities. This was introduced for SGD in Smith & Le (2018); Smith et al. (2018), and we now generalize it to our broader setting. For gradient descent on a loss $\mathcal{L}$, the intrinsic time is a quantity which increments by $\eta \lambda_{\min}(H)$ for a optimization step with learning rate $\eta$ at a point where $\mathcal{L}$ has Hessian $H$. When specialized to our setting, it is given by

$$\tau(t) := \sum_{s=0}^{t-1} \frac{2\eta_s}{N} \lambda_{\min, V_\parallel}(\mathbb{E}[X_s X_s^\mathsf{T}]). \tag{5.12}$$

Notice that intrinsic time of augmented optimization for the sequence of proxy losses $\overline{\mathcal{L}}_s$ appears in Theorems 5.1 and 5.2, which require via condition (5.6) that the intrinsic time tends to infinity as the number of optimization steps grows.

Intrinsic time will be a sensible variable in which to measure the behavior of quantities such as the fluctuations of the optimization path $f(t) := \mathbb{E}[\|(W_t - \mathbb{E}[W_t])Q_\parallel\|_F^2]$. In the proofs of Theorems 5.1 and 5.2, we show that the fluctuations satisfy an inequality of the form

$$f(t+1) \le f(t)(1 - a(t))^2 + b(t) \tag{5.13}$$

for $a(t) := 2\eta_t \frac{1}{N} \lambda_{\min, V_\parallel}(\mathbb{E}[X_t X_t^\mathsf{T}])$ and $b(t) := \mathrm{Var}[\|\eta_t \nabla_W \mathcal{L}(W_t)\|_F]$ so that $\tau(t) = \sum_{s=0}^{t-1} a(s)$. Iterating the recursion (5.13) shows that

$$f(t) \le f(0) \prod_{s=0}^{t-1}(1 - a(s))^2 + \sum_{s=0}^{t-1} b(s) \prod_{r=s+1}^{t-1}(1 - a(r))^2$$

$$\le e^{-2\tau(t)} f(0) + \sum_{s=0}^{t-1} \frac{b(s)}{a(s)} e^{2\tau(s+1) - 2\tau(t)}(\tau(s+1) - \tau(s)).$$

For $\tau := \tau(t)$ and changes of variable $A(\tau)$, $B(\tau)$, and $F(\tau)$ such that $A(\tau(t)) = a(t)$, $B(\tau(t)) = b(t)$, and $F(\tau(t)) = f(t)$, we find by replacing a right Riemann sum by an integral that

$$F(\tau) \precsim e^{-2\tau} \left[ F(0) + \int_0^\tau \frac{B(\sigma)}{A(\sigma)} e^{2\sigma} d\sigma \right]. \tag{5.14}$$

In order for the result of optimization to be independent of the starting point, by (5.14) we must have $\tau \to \infty$ to remove the dependence on $F(0)$; this provides one explanation for the appearance of $\tau$ in condition (5.6). Further, (5.14) implies that the fluctuations at an intrinsic time are bounded by an integral against the function $\frac{B(\sigma)}{A(\sigma)}$ which depends only on the ratio of $A(\sigma)$ and $B(\sigma)$. In the case of minibatch SGD, we compute this ratio in (6.2) and recover the commonly used "linear scaling" rule for learning rate.

In Section 6, we specialize Theorem 5.2 to obtain rates of convergence for specific augmentations. Optimizing the learning rate and augmentation parameter schedules in Theorem 5.2 allows us to derive power law schedules with convergence rate guarantees in these settings.

# 6    IMPLICATIONS FOR MINI-BATCH STOCHASTIC GRADIENT DESCENT (SGD)

We now apply our framework to study mini-batch stochastic gradient descent (SGD) with the potential presence of additive noise. Though data augmentation commonly refers to techniques aside from SGD, we will see that our framework handles it uniformly with other augmentations.

## 6.1    MINI-BATCH SGD

In mini-batch stochastic gradient descent, $\mathcal{D}_t$ is obtained by choosing a random subset $\mathcal{B}_t$ of $\mathcal{D}$ of prescribed batch size $B_t = |\mathcal{B}_t|$. Each datapoint in $\mathcal{B}_t$ is chosen uniformly with replacement from $\mathcal{D}$, and the resulting data matrices $X_t$ and $Y_t$ are scaled so that $\overline{\mathcal{L}}_t(W) = \mathcal{L}(W; \mathcal{D})$. Concretely, this means that for the normalizing factor $c_t := \sqrt{N/B_t}$ we have

$$X_t = c_t X A_t \qquad \text{and} \qquad Y_t = c_t Y A_t,$$

where $A_t \in \mathbb{R}^{N \times B_t}$ has i.i.d. columns $A_{t,i}$ with a single non-zero entry equal to 1 chosen uniformly at random. In this setting the minimum norm optimum for each $t$ are the same and given by

$$W_t^* = W_\infty^* = Y X^\mathsf{T} (X X^\mathsf{T})^+,$$

which coincides with the minimum norm optimum for the unaugmented loss. Our main result for standard SGD is the following theorem, whose proof is given in Appendix D.1.

**Theorem 6.1.** *If the learning rate satisfies $\eta_t \to 0$ and*

$$\sum_{t=0}^{\infty} \eta_t = \infty, \tag{6.1}$$

*then for any initialization $W_0$, we have $W_t Q_\parallel \xrightarrow{p} W_\infty^*$. If further we have that $\eta_t = \Theta(t^{-x})$ with $0 < x < 1$, then for some $C > 0$ we have*

$$e^{C t^{1-x}} \|W_t Q_\parallel - W_\infty^*\|_F \xrightarrow{p} 0.$$

Theorem 6.1 recovers the exponential convergence rate for SGD, which has been extensively studied through both empirical and theoretical means (Bottou et al., 2018; Ma et al., 2018). Because $1 \leq B_t \leq N$ for all $t$, it does not affect the asymptotic results in Theorem 6.1. In practice, however, the number of optimization steps $t$ is often small enough that $\frac{B_t}{N}$ is of order $t^{-\alpha}$ for some $\alpha > 0$, meaning the choice of $B_t$ can affect rates in this non-asymptotic regime. Though we do not attempt to push our generic analysis to this granularity, this is done in Ma et al. (2018) to derive optimal batch sizes and learning rates in the overparametrized setting.

Our proof of Theorem 6.1 shows the intrinsic time is $\tau(t) = \sum_{s=0}^{t-1} 2\eta_s \frac{1}{N} \lambda_{\min, V_\parallel}(X X^\mathsf{T})$ and the ratio $\frac{b(t)}{a(t)}$ in (5.14) is by (D.4) bounded uniformly for a constant $C > 0$ by

$$\frac{b(t)}{a(t)} \leq C \cdot \frac{\eta_t}{B_t}. \tag{6.2}$$

Thus, keeping $\frac{b(t)}{a(t)}$ fixed as a function of $\tau$ suggests the "linear scaling" $\eta_t \propto B_t$ used empirically in Goyal et al. (2017) and proposed via an heuristic SDE limit in Smith et al. (2018).

## 6.2 MINI-BATCH SGD WITH ADDITIVE SYNTHETIC NOISE

In addition to handling synthetic noise and SGD separately, our results and framework also cover the hybrid case of mini-batch SGD with batch size $B_t$ and additive noise at level $\sigma_t$. Here,

$$X_t = c_t(XA_t + \sigma_t G_t) \qquad \text{and} \qquad Y_t = c_t Y A_t,$$

where $c_t$ and $A_t$ are as in Section 6.1 and $G_t \in \mathbb{R}^{n \times B_t}$ has i.i.d. Gaussian entries. The proxy loss is

$$\overline{\mathcal{L}}_t(W) := \frac{1}{N}\mathbb{E}\left[\|c_t Y A_t - c_t W X A_t - c_t \sigma_t W G_t\|_F^2\right] = \frac{1}{N}\|Y - WX\|_F^2 + \sigma_t^2\|W\|_F^2,$$

with ridge minimizer $W_t^* = YX^\mathsf{T}(XX^\mathsf{T} + \sigma_t^2 N \cdot \mathrm{Id}_{n \times n})^{-1}$. Like with synthetic noise but unlike noiseless SGD, the optima $W_t^*$ converge to the minimal norm interpolant $W_{\min} = YX^\mathsf{T}(XX^\mathsf{T})^+$.

**Theorem 6.2.** *Suppose $\sigma_t^2 \to 0$ is decreasing, $\eta_t \to 0$, and for any $C > 0$ we have*

$$\sum_{t=0}^{\infty}(\eta_t \sigma_t^2 - C\eta_t^2) = \infty \qquad \text{and} \qquad \sum_{t=0}^{\infty} \eta_t^2 \sigma_t^2 < \infty. \tag{6.3}$$

*Then we have $W_t \xrightarrow{p} W_{min}$. If we further have $\eta_t = \Theta(t^{-x})$ and $\sigma_t^2 = \Theta(t^{-y})$ with $x, y > 0$ and $0 < x + y < 1 < 2x + y$, we have for any $\epsilon > 0$ that*

$$t^{\min\{y, \frac{1}{2}x\} - \epsilon}\|W_t - W_{min}\|_F \xrightarrow{p} 0.$$

Theorem 6.2 provides an example where our framework can handle the *composition* of two augmentations, namely additive noise and SGD. It reveals a qualitative difference between SGD with and without additive noise. For polynomially decaying $\eta_t$ the convergence of noiseless SGD in Theorem 6.1 is exponential in $t$, while the bound from Theorem 6.2 is polynomial in $t$. This is unavoidable. Indeed, for components of $W_t$ orthogonal to $\mathrm{colspan}(X)$, convergence requires that $\sum_{t=0}^{\infty} \eta_t \sigma_t^2 = \infty$ (see (4.3)). This occurs only if $\sigma_t$ has power law decay, causing the $\|W_t^* - W_{\min}\|_F$ to have at most power law decay as well. Finally, the Monro-Robbins conditions (6.3) are more restrictive than the analogous conditions in the pure noise setting (see (4.4)), as the latter allow for large $\eta_t$ schedules in which $\sum_{t=0}^{\infty} \eta_t^2$ diverges but $\sum_{t=0}^{\infty} \eta_t^2 \sigma_t^2$ does not.

## 7 DISCUSSION

We have presented a theoretical framework to rigorously analyze the effect of data augmentation. As can be seen in our main results, our framework applies to completely general augmentations and relies only on analyzing the first few moments of the augmented dataset. This allows us to handle augmentations as diverse as additive noise and mini-batch SGD as well as their composition in a uniform manner. We have analyzed some representative examples in detail in this work, but many other commonly used augmentations may be handled similarly: label-preserving transformations (e.g. color jitter, geometric transformations), random projections (DeVries & Taylor, 2017; Park et al., 2019), and Mixup (Zhang et al., 2017), among many others. Another line of investigation left to future work is to compare different methods of combining augmentations such as mixing, alternating, or composing, which often improve performance in the empirical literature (Hendrycks et al., 2020).

Though our results provide a rigorous baseline to compare to more complex settings, the restriction of the present work to linear models is of course a significant constraint. In future work, we hope to extend our general analysis to models closer to those used in practice. Most importantly, we intend to consider more complex models such as kernels (including the neural tangent kernel) and neural networks by making similar connections to stochastic optimization. In an orthogonal direction, our analysis currently focuses on the mean square loss for regression, and we aim to extend it to other losses such as the cross-entropy loss. Finally, our study has thus far been restricted to the effect of data augmentation on optimization, and it would be of interest to derive consequences for generalization with more complex models. We hope our framework can provide the theoretical underpinnings for a more principled understanding of the effect and practice of data augmentation.

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

## A  ANALYTIC LEMMAS

In this section, we present several basic lemmas concerning convergence for certain matrix-valued recursions that will be needed to establish our main results. For clarity, we first collect some matrix notations used in this section and throughout the paper.

### A.1  MATRIX NOTATIONS

Let $M \in \mathbb{R}^{m \times n}$ be a matrix. We denote its Frobenius norm by $\|M\|_F$ and its spectral norm by $\|M\|_2$. If $m = n$ so that $M$ is square, we denote by $\mathrm{diag}(M)$ the diagonal matrix with $\mathrm{diag}(M)_{ii} = M_{ii}$. For matrices $A, B, C$ of the appropriate shapes, define

$$A \circ (B \otimes C) := BAC \tag{A.1}$$

and

$$\mathrm{Var}(A) := \mathbb{E}[A^\mathsf{T} \otimes A] - \mathbb{E}[A^\mathsf{T}] \otimes \mathbb{E}[A]. \tag{A.2}$$

Notice in particular that

$$\mathrm{Tr}[\mathrm{Id} \circ \mathrm{Var}(A)] = \mathbb{E}[\|A - \mathbb{E}[A]\|_F^2].$$

## A.2 ONE- AND TWO-SIDED DECAY

**Definition A.1.** *Let $A_t \in \mathbb{R}^{n \times n}$ be a sequence of independent random non-negative definite matrices with*

$$\sup_t \|A_t\| \le 2 \quad almost\ surely,$$

*let $B_t \in \mathbb{R}^{p \times n}$ be a sequence of arbitrary matrices, and let $C_t \in \mathbb{R}^{n \times n}$ be a sequence of non-negative definite matrices. We say that the sequence of matrices $X_t \in \mathbb{R}^{p \times n}$ has one-sided decay of type $(\{A_t\}, \{B_t\})$ if it satisfies*

$$X_{t+1} = X_t(\mathrm{Id} - \mathbb{E}[A_t]) + B_t. \tag{A.3}$$

*We say that a sequence of non-negative definite matrices $Z_t \in \mathbb{R}^{n \times n}$ has two-sided decay of type $(\{A_t\}, \{C_t\})$ if it satisfies*

$$Z_{t+1} = \mathbb{E}[(\mathrm{Id} - A_t)Z_t(\mathrm{Id} - A_t)] + C_t. \tag{A.4}$$

Intuitively, if a sequence of matrices $X_t$ (resp. $Z_t$) satisfies one decay of type $(\{A_t\}, \{B_t\})$ (resp. two-sided decay of type $(\{A_t\}, \{C_t\})$), then in those directions $u \in \mathbb{R}^n$ for which $\|A_t u\|$ does not decay too quickly in $t$ we expect that $X_t$ (resp. $Z_t$) will converge to 0 provided $B_t$ (resp. $C_t$) are not too large. More formally, let us define

$$V_{\|} := \bigcap_{t=0}^{\infty} \ker \left[ \prod_{s=t}^{\infty} (\mathrm{Id} - \mathbb{E}[A_s]) \right] = \left\{ u \in \mathbb{R}^n \ \middle| \ \lim_{T \to \infty} \prod_{s=t}^{T} (\mathrm{Id} - \mathbb{E}[A_s]) u = 0, \quad \forall t \ge 1 \right\},$$

and let $Q_{\|}$ be the orthogonal projection onto $V_{\|}$. It is on the space $V_{\|}$ that that we expect $X_t, Z_t$ to tend to zero if they satisfy one or two-side decay, and the precise results follows.

## A.3 LEMMAS ON CONVERGENCE FOR MATRICES WITH ONE AND TWO-SIDED DECAY

We state here several results that underpin the proofs of our main results. We begin by giving in Lemmas A.2 and A.3 two slight variations of the same simple argument that matrices with one or two-sided decay converge to zero.

**Lemma A.2.** *If a sequence $\{X_t\}$ has one-sided decay of type $(\{A_t\}, \{B_t\})$ with*

$$\sum_{t=0}^{\infty} \|B_t\|_F < \infty, \tag{A.5}$$

*then $\lim_{t \to \infty} X_t Q_{\|} = 0$.*

*Proof.* For any $\epsilon > 0$, choose $T_1$ so that $\sum_{t=T_1}^{\infty} \|B_t\|_F < \frac{\epsilon}{2}$ and $T_2$ so that for $t > T_2$ we have

$$\left\| \left( \prod_{s=T_1}^{t} (\mathrm{Id} - \mathbb{E}[A_s]) \right) Q_{\|} \right\|_2 < \frac{\epsilon}{2} \frac{1}{\|X_0\|_F + \sum_{s=0}^{T_1-1} \|B_s\|_F}.$$

By (A.3), we find that

$$X_{t+1} = X_0 \prod_{s=0}^{t} (\mathrm{Id} - \mathbb{E}[A_s]) + \sum_{s=0}^{t} B_s \prod_{r=s+1}^{t} (\mathrm{Id} - \mathbb{E}[A_r]),$$

which implies for $t > T_2$ that

$$\|X_{t+1} Q_{\|}\|_F \le \|X_0\|_F \left\| \left( \prod_{s=0}^{t} (\mathrm{Id} - \mathbb{E}[A_s]) \right) Q_{\|} \right\|_2 + \sum_{s=0}^{t} \|B_s\|_F \left\| \left( \prod_{r=s+1}^{t} (\mathrm{Id} - \mathbb{E}[A_r]) \right) Q_{\|} \right\|_2. \tag{A.6}$$

Our assumption that $\|A_t\| \le 2$ almost surely implies that for any $T \le t$

$$\left\| \left( \prod_{s=0}^{t} (\mathrm{Id} - \mathbb{E}[A_s]) \right) Q_{\|} \right\|_2 \le \left\| \left( \prod_{s=0}^{T} (\mathrm{Id} - \mathbb{E}[A_s]) \right) Q_{\|} \right\|_2$$

since each term in the product is non-negative-definite. Thus, we find

$$\|X_{t+1}Q_\|\|_F \leq \left[\|X_0\|_F + \sum_{s=0}^{T_1-1}\|B_s\|_F\right]\left\|\left(\prod_{s=T_1}^{t}(\mathrm{Id} - \mathbb{E}[A_s])\right)Q_\|\right\|_2 + \sum_{s=T_1}^{t}\|B_s\|_F < \epsilon.$$

Taking $t \to \infty$ and then $\epsilon \to 0$ implies that $\lim_{t\to\infty} X_t Q_\| = 0$, as desired. $\square$

**Lemma A.3.** *If a sequence $\{Z_t\}$ has two-sided decay of type $(\{A_t\}, \{C_t\})$ with*

$$\lim_{T\to\infty} \mathbb{E}\left[\left\|\left(\prod_{s=t}^{T}(\mathrm{Id} - A_s)\right)Q_\|\right\|_2^2\right] = 0 \quad \text{for all } t \geq 0 \tag{A.7}$$

*and*

$$\sum_{t=0}^{\infty} \mathrm{Tr}(C_t) < \infty, \tag{A.8}$$

*then $\lim_{t\to\infty} Q_\|^\mathsf{T} Z_t Q_\| = 0$.*

*Proof.* The proof is essentially identical to that of Lemma A.2. That is, for $\epsilon > 0$, choose $T_1$ so that $\sum_{t=T_1}^{\infty} \mathrm{Tr}(C_t) < \frac{\epsilon}{2}$ and choose $T_2$ by (A.7) so that for $t > T_2$ we have

$$\mathbb{E}\left[\left\|\left(\prod_{s=T_1}^{t}(\mathrm{Id} - A_s)\right)Q_\|\right\|_2^2\right] < \frac{\epsilon}{2}\frac{1}{\mathrm{Tr}(Z_0) + \sum_{s=0}^{T_1-1}\mathrm{Tr}(C_s)}.$$

Conjugating (A.4) by $Q_\|$, we have that

$$Q_\|^\mathsf{T} Z_{t+1} Q_\| = \mathbb{E}\left[Q_\|^\mathsf{T}\left(\prod_{s=0}^{t}(\mathrm{Id} - A_s)\right)^\mathsf{T} Z_0\left(\prod_{s=0}^{t}(\mathrm{Id} - A_s)\right)Q_\|\right]$$
$$+ \sum_{s=0}^{t}\mathbb{E}\left[Q_\|^\mathsf{T}\left(\prod_{r=s+1}^{t}(\mathrm{Id} - A_r)\right)^\mathsf{T} C_s\left(\prod_{r=s+1}^{t}(\mathrm{Id} - A_r)\right)Q_\|\right].$$

Our assumption that $\|A_t\| \leq 2$ almost surely implies that for any $T \leq t$

$$\left\|\left(\prod_{s=0}^{t}(\mathrm{Id} - A_s)\right)Q\right\|_2 \leq \left\|\left(\prod_{s=0}^{T}(\mathrm{Id} - A_s)\right)Q\right\|_2.$$

For $t > T_2$, this implies by taking trace of both sides that

$$\mathrm{Tr}(Q_\|^\mathsf{T} Z_{t+1} Q_\|) \leq \mathrm{Tr}(Z_0)\mathbb{E}\left[\left\|\left(\prod_{s=0}^{t}(\mathrm{Id} - A_s)\right)Q_\|\right\|_2^2\right] + \sum_{s=0}^{t}\mathrm{Tr}(C_s)\mathbb{E}\left[\left\|\left(\prod_{r=s+1}^{t}(\mathrm{Id} - A_r)\right)Q_\|\right\|_2^2\right]$$

$$\tag{A.9}$$

$$\leq \left[\mathrm{Tr}(Z_0) + \sum_{s=0}^{T_1-1}\mathrm{Tr}(C_s)\right]\mathbb{E}\left[\left\|\left(\prod_{s=T_1}^{t}(\mathrm{Id} - A_s)\right)Q_\|\right\|_2^2\right] + \sum_{s=T_1}^{t}\mathrm{Tr}(C_s)$$

$$< \epsilon,$$

which implies that $\lim_{t\to\infty} Q_\|^\mathsf{T} Z_t Q_\| = 0$. $\square$

The preceding Lemmas will be used to provide sufficient conditions for augmented gradient descent to converge as in Theorem B.1 below. Since we are also interested in obtaining rates of convergence, we record here two quantitative refinements of the Lemmas above that will be used in the proof of Theorem B.4.

**Lemma A.4.** *Suppose $\{X_t\}$ has one-sided decay of type $(\{A_t\}, \{B_t\})$. Assume also that for some $X \geq 0$ and $C > 0$, we have*

$$\log \left\| \left( \prod_{r=s}^{t} (\mathrm{Id} - \mathbb{E}[A_r]) \right) Q_{\|} \right\|_2 < X - C \int_s^{t+1} r^{-\alpha} dr$$

*and $\|B_t\|_F = O(t^{-\beta})$ for some $0 < \alpha < 1 < \beta$. Then, $\|X_t Q_{\|}\|_F = O(t^{\alpha - \beta})$.*

*Proof.* Denote $\gamma_{s,t} := \int_s^t r^{-\alpha} dr$. By (A.6), we have for some constants $C_1, C_2 > 0$ that

$$\|X_{t+1} Q_{\|}\|_F < C_1 e^{-C\gamma_{1,t+1}} + C_2 e^X \sum_{s=1}^{t} (1+s)^{-\beta} e^{-C\gamma_{s+1,t+1}}. \tag{A.10}$$

The first term on the right hand side is exponentially decaying in $t$ since $\gamma_{1,t+1}$ grows polynomially in $t$. To bound the second term, observe that the function

$$f(s) := C\gamma_{s+1,t+1} - \beta \log(s+1)$$

satisfies

$$f'(s) \geq 0 \quad \Leftrightarrow \quad C(s+1)^{-\alpha} - \frac{\beta}{1+s} \geq 0 \quad \Leftrightarrow \quad s \geq \left(\frac{\beta}{C}\right)^{1/(1-\alpha)} =: K.$$

Hence, the summands are monotonically increasing for $s$ greater than a fixed constant $K$ depending only on $\alpha, \beta, C$. Note that

$$\sum_{s=1}^{K} (1+s)^{-\beta} e^{-C\gamma_{s+1,t+1}} \leq K e^{-C\gamma_{K+1,t+1}} \leq K e^{-C' t^{1-\alpha}}$$

for some $C'$ depending only on $\alpha$ and $K$, and hence sum is exponentially decaying in $t$. Further, using an integral comparison, we find

$$\sum_{s=K+1}^{t} (1+s)^{-\beta} e^{-C\gamma_{s+1,t+1}} \leq \int_K^t (1+s)^{-\beta} e^{-\frac{C}{1-\alpha}\left((t+1)^{1-\alpha} - (s+1)^{1-\alpha}\right)} ds. \tag{A.11}$$

Changing variables using $u = (1+s)^{1-\alpha}/(1-\alpha)$, the last integral has the form

$$e^{-Cg_t} (1-\alpha)^{-\xi} \int_{g_K}^{g_t} u^{-\xi} e^{Cu} du, \qquad g_x := \frac{(1+x)^{1-\alpha}}{1-\alpha}, \, \xi := \frac{\beta - \alpha}{1 - \alpha}. \tag{A.12}$$

Integrating by parts, we have

$$\int_{g_K}^{g_t} u^{-\xi} e^u du = C^{-1} \xi \int_{g_K}^{g_t} u^{-\xi-1} e^{Cu} du + (u^{-\xi} e^{Cu})|_{g_K}^{g_t}$$

Further, since on the range $g_K \leq u \leq g_t$ the integrand is increasing, we have

$$e^{-Cg_t} \xi \int_{g_K}^{g_t} u^{-\xi-1} e^{Cu} du \leq \xi g_t^{-\xi}.$$

Hence, $e^{-Cg_t}$ times the integral in (A.12) is bounded above by

$$O(g_t^{-\xi}) + e^{-Cg_t} (u^{-\xi} e^{Cu})|_{g_K}^{g_t} = O(g_t^{-\xi}).$$

Using (A.11) and substituting the previous line into (A.12) yields the estimate

$$\sum_{s=K+1}^{t} (1+s)^{-\beta} e^{-C\gamma_{s+1,t+1}} \leq (1+t)^{-\beta+\alpha},$$

which completes the proof. □

**Lemma A.5.** *Suppose $\{Z_t\}$ has two-sided decay of type $(\{A_t\}, \{C_t\})$. Assume also that for some $X \geq 0$ and $C > 0$, we have*

$$\log \mathbb{E}\left[\left\|\left(\prod_{r=s}^{t}(\mathrm{Id} - A_r)\right)Q_\|\right\|_2^2\right] < X - C\int_s^{t+1} r^{-\alpha}dr$$

*as well as $\mathrm{Tr}(C_t) = O(t^{-\beta})$ for some $0 < \alpha < 1 < \beta$. Then $\mathrm{Tr}(Q_\|^T Z_t Q_\|) = O(t^{\alpha-\beta})$.*

*Proof.* This argument is identical to the proof of Lemma A.4. Indeed, using (A.9) we have that

$$\mathrm{Tr}\left(Q_\|^T Z_t Q_\|\right) \leq C_1 e^{-C\gamma_{1,t+1}} + C_2 e^X \sum_{s=1}^{t}(1+s)^{-\beta}e^{-C\gamma_{s+1,t+1}}.$$

The right hand side of this inequality coincides with the expression on the right hand side of (A.10), which we already bounded by $O(t^{\beta-\alpha})$ in the proof of Lemma A.4. $\quad\square$

In what follows, we will use a concentration result for products of matrices from Huang et al. (2020). Let $Y_1, \ldots, Y_n \in \mathbb{R}^{N \times N}$ be independent random matrices. Suppose that

$$\|\mathbb{E}[Y_i]\|_2 \leq a_i \qquad \text{and} \qquad \mathbb{E}\left[\|Y_i - \mathbb{E}[Y_i]\|_2^2\right] \leq b_i^2 a_i^2$$

for some $a_1, \ldots, a_n$ and $b_1, \ldots, b_n$. We will use the following result, which is a specialization of (Huang et al., 2020, Theorem 5.1) for $p = q = 2$.

**Theorem A.6** ((Huang et al., 2020, Theorem 5.1)). *For $Z_0 \in \mathbb{R}^{N \times n}$, the product $Z_n = Y_n Y_{n-1} \cdots Y_1 Z_0$ satisfies*

$$\mathbb{E}\left[\|Z_n\|_2^2\right] \leq e^{\sum_{i=1}^{n} b_i^2}\prod_{i=1}^{n} a_i^2 \cdot \|Z_0\|_2^2$$

$$\mathbb{E}\left[\|Z_n - \mathbb{E}[Z_n]\|_2^2\right] \leq \left(e^{\sum_{i=1}^{n} b_i^2} - 1\right)a_i^2 \cdot \|Z_0\|_2^2.$$

Finally, we collect two simple analytic lemmas for later use.

**Lemma A.7.** *For any matrix $M \in \mathbb{R}^{m \times n}$, we have that*

$$\mathbb{E}[\|M\|_2^2] \geq \|\mathbb{E}[M]\|_2^2.$$

*Proof.* We find by Cauchy-Schwartz and the convexity of the spectral norm that

$$\mathbb{E}[\|M\|_2^2] \geq \mathbb{E}[\|M\|_2]^2 \geq \|\mathbb{E}[M]\|_2^2. \qquad\square$$

**Lemma A.8.** *For bounded $a_t \geq 0$, if we have $\sum_{t=0}^{\infty} a_t = \infty$, then for any $C > 0$ we have*

$$\sum_{t=0}^{\infty} a_t e^{-C\sum_{s=0}^{t} a_s} < \infty.$$

*Proof.* Define $b_t := \sum_{s=0}^{t} a_s$ so that

$$S := \sum_{t=0}^{\infty} a_t e^{-C\sum_{s=0}^{t} a_s} = \sum_{t=0}^{\infty}(b_t - b_{t-1})e^{-Cb_t} \leq \int_0^{\infty} e^{-Cx}dx < \infty,$$

where we use $\int_0^{\infty} e^{-Cx}dx$ to upper bound its right Riemann sum. $\quad\square$

# B  ANALYSIS OF DATA AUGMENTATION AS STOCHASTIC OPTIMIZATION

In this section, we prove generalizations of our main theoretical results Theorems 5.1 and 5.2 giving Monro-Robbins type conditions for convergence and rates for augmented gradient descent in the linear setting.

## B.1 Monro-Robbins type results

To state our general Monro-Robbins type convergence results, let us briefly recall the notation. We consider overparameterized linear regression with loss

$$\mathcal{L}(W; \mathcal{D}) = \frac{1}{N} \|WX - Y\|_F^2,$$

where the dataset $\mathcal{D}$ of size $N$ consists of data matrices $X, Y$ that each have $N$ columns $x_i \in \mathbb{R}^n, y_i \in \mathbb{R}^p$ with $n > N$. We optimize $\mathcal{L}(W; \mathcal{D})$ by augmented gradient descent, which means that at each time $t$ we replace $\mathcal{D} = (X, Y)$ by a random dataset $\mathcal{D}_t = (X_t, Y_t)$. We then take a step

$$W_{t+1} = W_t - \eta_t \nabla_W \mathcal{L}(W_t; \mathcal{D}_t)$$

of gradient descent on the resulting randomly augmented loss $\mathcal{L}(W; \mathcal{D}_t)$ with learning rate $\eta_t$. Recall that we set

$$V_\| := \text{ column span of } \mathbb{E}[X_t X_t^\mathsf{T}]$$

and denoted by $Q_\|$ the orthogonal projection onto $V_\|$. As noted in §5, on $V_\|$ the proxy loss

$$\overline{\mathcal{L}}_t = \mathbb{E}\left[\mathcal{L}(W; \mathcal{D}_t)\right]$$

is strictly convex and has a unique minimum, which is

$$W_t^* = \mathbb{E}\left[Y_t X_t^T\right] (Q_\| \mathbb{E}\left[X_t X_t^\mathsf{T}\right] Q_\|)^{-1}.$$

The change from one step of augmented GD to the next in these proxy optima is captured by

$$\Xi_t^* := W_{t+1}^* - W_t^*.$$

With this notation, we are ready to state Theorems B.1, which gives two different sets of time-varying Monro-Robbins type conditions under which the optimization trajectory $W_t$ converges for large $t$. In Theorem B.4, we refine the analysis to additionally give rates of convergence.

**Theorem B.1.** *Suppose that $V_\|$ is independent of $t$, that the learning rate satisfies $\eta_t \to 0$, that the proxy optima satisfy*

$$\sum_{t=0}^\infty \|\Xi_t^*\|_F < \infty, \tag{B.1}$$

*ensuring the existence of a limit $W_\infty^* := \lim_{t\to\infty} W_t^*$ and that*

$$\sum_{t=0}^\infty \eta_t \lambda_{min, V_\|} (\mathbb{E}[X_t X_t^\mathsf{T}]) = \infty. \tag{B.2}$$

*Then if either*

$$\sum_{t=0}^\infty \eta_t^2 \mathbb{E}\left[\|X_t X_t^\mathsf{T} - \mathbb{E}[X_t X_t^\mathsf{T}]\|_F^2 + \|Y_t X_t^\mathsf{T} - \mathbb{E}[Y_t X_t^\mathsf{T}]\|_F^2\right] < \infty \tag{B.3}$$

*or*

$$\sum_{t=0}^\infty \eta_t^2 \mathbb{E}\Big[\|X_t X_t^\mathsf{T} - \mathbb{E}[X_t X_t^\mathsf{T}]\|_F^2$$

$$+ \left\|\mathbb{E}[W_t](X_t X_t^\mathsf{T} - \mathbb{E}[X_t X_t^\mathsf{T}]) - (Y_t X_t^\mathsf{T} - \mathbb{E}[Y_t X_t^\mathsf{T}])\right\|_F^2\Big] < \infty \tag{B.4}$$

*hold, then for any initialization $W_0$, we have $W_t Q_\| \xrightarrow{p} W_\infty^*$.*

**Remark B.2.** *In the general case, the column span $V_\|$ of $\mathbb{E}[X_t X_t^\mathsf{T}]$ may vary with $t$. This means that some directions in $\mathbb{R}^n$ may only have non-zero overlap with $\text{colspan}(\mathbb{E}[X_t X_t^\mathsf{T}])$ for some positive but finite collection of values of $t$. In this case, only finitely many steps of the optimization would move $W_t$ in this direction, meaning that we must define a smaller space for convergence. The correct definition of this subspace turns out to be the following*

$$V_\| := \bigcap_{t=0}^\infty \ker\left[\prod_{s=t}^\infty \left(\text{Id} - \frac{2\eta_s}{N} \mathbb{E}[X_s X_s^\mathsf{T}]\right)\right] \tag{B.5}$$

$$= \bigcap_{t=0}^\infty \left\{u \in \mathbb{R}^n \ \Big|\ \lim_{T\to\infty} \prod_{s=t}^T \left(\text{Id} - \frac{2\eta_s}{N} \mathbb{E}[X_s X_s^\mathsf{T}]\right) u = 0\right\}.$$

*With this re-definition of $V_\parallel$ and with $Q_\parallel$ still denoting the orthogonal projection to $V_\parallel$, Theorem B.1 holds verbatim and with the same proof. Note that if $\eta_t \to 0$, $V_\parallel \mathrm{colspan}(\mathbb{E}[X_t X_t^\mathsf{T}])$ is fixed in $t$, and (B.2) holds, this definition of $V_\parallel$ reduces to that defined in (5.3).*

**Remark B.3.** *The condition* (B.4) *can be written in a more conceptual way as*

$$\sum_{t=0}^\infty \left[ \|X_t X_t^\mathsf{T} - \mathbb{E}[X_t X_t^\mathsf{T}]\|_F^2 + \eta_t^2 \operatorname{Tr}\left[ \mathrm{Id} \circ \mathrm{Var}\left( (\mathbb{E}[W_t]X_t - Y_t)X_t^\mathsf{T} \right) \right] \right] < \infty,$$

*where we recognize that $(\mathbb{E}[W_t]X_t - Y_t)X_t^\mathsf{T}$ is precisely the stochastic gradient estimate at time $t$ for the proxy loss $\overline{\mathcal{L}}_t$, evaluated at $\mathbb{E}[W_t]$, which is the location at time $t$ for vanilla GD on $\overline{\mathcal{L}}_t$ since taking expectations in the GD update equation (5.1) coincides with GD for $\overline{\mathcal{L}}_t$. Moreover, condition* (B.4) *actually implies condition* (B.3) *(see* (B.12) *below). The reason we state Theorem B.1 with both conditions, however, is that* (B.4) *makes explicit reference to the average $\mathbb{E}[W_t]$ of the augmented trajectory. Thus, when applying Theorem B.1 with this weaker condition, one must separately estimate the behavior of this quantity.*

Theorem B.1 gave conditions on joint learning rate and data augmentation schedules under which augmented optimization is guaranteed to converge. Our next result proves rates for this convergence.

**Theorem B.4.** *Suppose that $\eta_t \to 0$ and that for some $0 < \alpha < 1 < \beta_1, \beta_2$ and $C_1, C_2 > 0$, we have*

$$\log \mathbb{E}\left[ \left\| \left( \prod_{r=s}^t \left( \mathrm{Id} - \frac{2\eta_r}{N} X_r X_r^\mathsf{T} \right) \right) Q_\parallel \right\|_2^2 \right] < C_1 - C_2 \int_s^{t+1} r^{-\alpha} dr \qquad \text{(B.6)}$$

*as well as*

$$\|\Xi_t^*\|_F = O(t^{-\beta_1}) \qquad \text{(B.7)}$$

*and*

$$\eta_t^2 \operatorname{Tr}\left[ \mathrm{Id} \circ \mathrm{Var}(\mathbb{E}[W_t]X_t X_t^\mathsf{T} - Y_t X_t^\mathsf{T}) \right] = O(t^{-\beta_2}). \qquad \text{(B.8)}$$

*Then, for any initialization $W_0$, we have for any $\epsilon > 0$ that*

$$t^{\min\{\beta_1 - 1, \frac{\beta_2 - \alpha}{2}\} - \epsilon} \|W_t Q_\parallel - W_\infty^*\|_F \xrightarrow{p} 0.$$

**Remark B.5.** *To reduce Theorem 5.2 to Theorem B.4, we notice that* (5.9) *and* (5.10) *mean that Theorem A.6 applies to $Y_t = \mathrm{Id} - 2\eta_t \frac{X_t X_t^\mathsf{T}}{N}$ with $a_t = 1 - \Omega(t^{-\alpha})$ and and $b_t^2 = O(t^{-\gamma})$, thus implying* (B.6).

The first step in proving both Theorem B.1 and Theorem B.4 is to obtain recursions for the mean and variance of the difference $W_t - W_t^*$ between the time $t$ proxy optimum and the augmented optimization trajectory at time $t$. We will then complete the proof of Theorem B.1 in §B.3 and the proof of Theorem B.4 in §B.4.

## B.2 Recursion relations for parameter moments

The following proposition shows that difference between the mean augmented dynamics $\mathbb{E}[W_t]$ and the time$-t$ optimum $W_t^*$ satisfies, in the sense of Definition A.1, one-sided decay of type $(\{A_t\}, \{B_t\})$ with

$$A_t = \frac{2\eta_t}{N} X_t X_t^\mathsf{T}, \qquad B_t = -\Xi_t^*.$$

It also shows that the variance of this difference, which is non-negative definite, satisfies two-sided decay of type $(\{A_t\}, \{C_t\})$ with $A_t$ as before and

$$C_t = \frac{4\eta_t^2}{N^2} \left[ \mathrm{Id} \circ \mathrm{Var}\left( \mathbb{E}[W_t]X_t X_t^\mathsf{T} - Y_t X_t^\mathsf{T} \right) \right].$$

In terms of the notations of Appendix A.1, we have the following recursions.

**Proposition B.6.** *The quantity* $\mathbb{E}[W_t] - W_t^*$ *satisfies*

$$\mathbb{E}[W_{t+1}] - W_{t+1}^* = (\mathbb{E}[W_t] - W_t^*)\Big(\mathrm{Id} - \frac{2\eta_t}{N}\mathbb{E}[X_t X_t^{\mathsf{T}}]\Big) - \Xi_t^* \tag{B.9}$$

*and* $Z_t := \mathbb{E}[(W_t - \mathbb{E}[W_t])^{\mathsf{T}}(W_t - \mathbb{E}[W_t])]$ *satisfies*

$$Z_{t+1} = \mathbb{E}\left[(\mathrm{Id} - \frac{2\eta_t}{N}X_t X_t^{\mathsf{T}})Z_t(\mathrm{Id} - \frac{2\eta_t}{N}X_t X_t^{\mathsf{T}})\right] + \frac{4\eta_t^2}{N^2}\left[\mathrm{Id}\circ\mathrm{Var}\Big(\mathbb{E}[W_t]X_t X_t^{\mathsf{T}} - Y_t X_t^{\mathsf{T}}\Big)\right]. \tag{B.10}$$

*Proof.* Notice that $\mathbb{E}[X_t X_t^{\mathsf{T}}]u = 0$ if and only if $X_t^{\mathsf{T}}u = 0$ almost surely, which implies that

$$W_t^*\mathbb{E}[X_t X_t^{\mathsf{T}}] = \mathbb{E}[Y_t X_t^{\mathsf{T}}]\mathbb{E}[X_t X_t^{\mathsf{T}}]^+\mathbb{E}[X_t X_t^{\mathsf{T}}] = \mathbb{E}[Y_t X_t^{\mathsf{T}}].$$

Thus, the learning dynamics (5.1) yield

$$\mathbb{E}[W_{t+1}] = \mathbb{E}[W_t] - \frac{2\eta_t}{N}\Big(\mathbb{E}[W_t]\mathbb{E}[X_t X_t^{\mathsf{T}}] - \mathbb{E}[Y_t X_t^{\mathsf{T}}]\Big)$$

$$= \mathbb{E}[W_t] - \frac{2\eta_t}{N}(\mathbb{E}[W_t] - W_t^*)\mathbb{E}[X_t X_t^{\mathsf{T}}].$$

Subtracting $W_{t+1}^*$ from both sides yields (B.9). We now analyze the fluctuations. Writing $\mathrm{Sym}(A) := A + A^{\mathsf{T}}$, we have

$$\mathbb{E}[W_{t+1}]^{\mathsf{T}}\mathbb{E}[W_{t+1}] = \mathbb{E}[W_t]^{\mathsf{T}}\mathbb{E}[W_t] + \frac{2\eta_t}{N}\mathrm{Sym}\Big(\mathbb{E}[W_t]^{\mathsf{T}}\mathbb{E}[Y_t X_t^{\mathsf{T}}] - \mathbb{E}[W_t]^{\mathsf{T}}\mathbb{E}[W_t]\mathbb{E}[X_t X_t^{\mathsf{T}}]\Big)$$

$$+ \frac{4\eta_t^2}{N^2}\Big(\mathbb{E}[X_t X_t^{\mathsf{T}}]\mathbb{E}[W_t]^{\mathsf{T}}\mathbb{E}[W_t]\mathbb{E}[X_t X_t^{\mathsf{T}}] + \mathbb{E}[X_t Y_t^{\mathsf{T}}]\mathbb{E}[Y_t X_t^{\mathsf{T}}] - \mathrm{Sym}(\mathbb{E}[X_t X_t^{\mathsf{T}}]\mathbb{E}[W_t]^{\mathsf{T}}\mathbb{E}[Y_t X_t^{\mathsf{T}}])\Big).$$

Similarly, we have that

$$\mathbb{E}[W_{t+1}^{\mathsf{T}}W_{t+1}] = \mathbb{E}[W_t^{\mathsf{T}}W_t] + \frac{2\eta_t}{N}\mathrm{Sym}(\mathbb{E}[W_t^{\mathsf{T}}Y_t X_t^{\mathsf{T}} - W_t^{\mathsf{T}}W_t X_t X_t^{\mathsf{T}}])$$

$$+ \frac{4\eta_t^2}{N^2}\mathbb{E}[X_t X_t^{\mathsf{T}}W_t^{\mathsf{T}}W_t X_t X_t^{\mathsf{T}} - \mathrm{Sym}(X_t X_t^{\mathsf{T}}W_t^{\mathsf{T}}Y_t X_t^{\mathsf{T}}) + X_t Y_t^{\mathsf{T}}Y_t X_t^{\mathsf{T}}].$$

Noting that $X_t$ and $Y_t$ are independent of $W_t$ and subtracting yields the desired. $\square$

### B.3 PROOF OF THEOREM B.1

First, by Proposition B.6, we see that $\mathbb{E}[W_t] - W_t^*$ has one-sided decay with

$$A_t = 2\eta_t\frac{X_t X_t^{\mathsf{T}}}{N} \qquad\text{and}\qquad B_t = -\Xi_t^*.$$

Thus, by Lemma A.2 and (B.1), we find that

$$\lim_{t\to\infty}(\mathbb{E}[W_t]Q_\| - W_t^*) = 0, \tag{B.11}$$

which gives convergence in expectation.

For the second moment, by Proposition B.6, we see that $Z_t$ has two-sided decay with

$$A_t = 2\eta_t\frac{X_t X_t^{\mathsf{T}}}{N} \qquad\text{and}\qquad C_t = \frac{4\eta_t^2}{N^2}\left[\mathrm{Id}\circ\mathrm{Var}\Big(\mathbb{E}[W_t]X_t X_t^{\mathsf{T}} - Y_t X_t^{\mathsf{T}}\Big)\right].$$

We now verify (A.7) and (A.8) in order to apply Lemma A.3.

For (A.7), for any $\epsilon > 0$, notice that

$$\mathbb{E}[\|A_s - \mathbb{E}[A_s]\|_F^2] = \eta_s^2\mathbb{E}[\|X_s X_s^{\mathsf{T}} - \mathbb{E}[X_s X_s^{\mathsf{T}}]\|_F^2]$$

so by either (B.3) or (B.4) we may choose $T_1 > t$ so that $\sum_{s=T_1}^\infty \mathbb{E}[\|A_s - \mathbb{E}[A_s]\|_F^2] < \frac{\epsilon}{2}$. Now choose $T_2 > T_1$ so that for $T > T_2$, we have

$$\left\|\Big(\prod_{r=T_1}^T \mathbb{E}[\mathrm{Id} - A_r]\Big)Q_\|\right\|_2^2 < \frac{\epsilon}{2}\frac{1}{\|\prod_{s=t}^{T_1-1}\mathbb{E}[\mathrm{Id} - A_s]\|_F^2 + \sum_{s=t}^{T_1-1}\mathbb{E}[\|A_s - \mathbb{E}[A_s]\|_F^2]}.$$

For $T > T_2$, we then have

$$\mathbb{E}\left[\left\|\left(\prod_{s=t}^{T}(\mathrm{Id}-A_s)\right)Q_\|\right\|_2^2\right]$$

$$\leq \left\|\left(\prod_{s=t}^{T}\mathbb{E}[\mathrm{Id}-A_s]\right)Q_\|\right\|^2 + \sum_{s=t}^{T}\mathbb{E}\left[\left\|\prod_{r=t}^{s}(\mathrm{Id}-A_r)\prod_{r=s+1}^{T}(\mathrm{Id}-\mathbb{E}[A_r])Q_\|\right\|_F^2 - \left\|\prod_{r=t}^{s-1}(\mathrm{Id}-A_r)\prod_{r=s}^{T}(\mathrm{Id}-\mathbb{E}[A_r])Q_\|\right\|_F^2\right]$$

$$= \left\|\left(\prod_{s=t}^{T}\mathbb{E}[\mathrm{Id}-A_s]\right)Q_\|\right\|_F^2 + \sum_{s=t}^{T}\mathbb{E}\left[\left\|\prod_{r=t}^{s-1}(\mathrm{Id}-A_r)(A_s-\mathbb{E}[A_s])\prod_{r=s+1}^{T}(\mathrm{Id}-\mathbb{E}[A_r])Q_\|\right\|_F^2\right]$$

$$\leq \left\|\prod_{s=t}^{T_1-1}\mathbb{E}[\mathrm{Id}-A_s]\right\|_F^2\left\|\left(\prod_{r=T_1}^{T}\mathbb{E}[\mathrm{Id}-A_r]\right)Q_\|\right\|_2^2 + \sum_{s=t}^{T}\mathbb{E}[\|A_s-\mathbb{E}[A_s]\|_F^2]\left\|\left(\prod_{r=s+1}^{T}\mathbb{E}[\mathrm{Id}-A_r]\right)Q_\|\right\|_2^2$$

$$\leq \left(\left\|\prod_{s=t}^{T_1-1}\mathbb{E}[\mathrm{Id}-A_s]\right\|_F^2 + \sum_{s=t}^{T_1-1}\mathbb{E}[\|A_s-\mathbb{E}[A_s]\|_F^2]\right)\left\|\left(\prod_{r=T_1}^{T}\mathbb{E}[\mathrm{Id}-A_r]\right)Q_\|\right\|_2^2 + \sum_{s=T_1}^{T}\mathbb{E}[\|A_s-\mathbb{E}[A_s]\|_F^2]$$

$$< \epsilon,$$

which implies (A.7). Condition (A.8) follows from either (B.4) or (B.3) and the bounds

$$\mathrm{Tr}(C_t) \leq \frac{8\eta_t^2}{N^2}\left(\|\mathbb{E}[W_t](X_tX_t^\mathsf{T}-\mathbb{E}[X_tX_t^\mathsf{T}])\|_F^2 + \|Y_tX_t^\mathsf{T}-\mathbb{E}[Y_tX_t^\mathsf{T}]\|_F^2\right) \tag{B.12}$$

$$\leq \frac{8\eta_t^2}{N^2}\left(\|\mathbb{E}[W_t]\|^2\|X_tX_t^\mathsf{T}-\mathbb{E}[X_tX_t^\mathsf{T}]\|_F^2 + \|Y_tX_t^\mathsf{T}-\mathbb{E}[Y_tX_t^\mathsf{T}]\|_F^2\right),$$

where in the first inequality we use the fact that $\|M_1-M_2\|_F^2 \leq 2(\|M_1\|_F^2+\|M_2\|_F^2)$. Furthermore, iterating (B.9) yields $\|\mathbb{E}[W_t]-W_t^*\|_F \leq \|W_0-W_0^*\|_F + \sum_{t=0}^{\infty}\|\Xi_t^*\|_F$, which combined with (B.12) and either (B.3) or (B.4) therefore implies (A.8). We conclude by Lemma A.3 that

$$\lim_{t\to\infty}Q_\|^\mathsf{T}Z_tQ_\| = \lim_{t\to\infty}\mathbb{E}[Q_\|^\mathsf{T}(W_t-\mathbb{E}[W_t])^\mathsf{T}(W_t-\mathbb{E}[W_t])Q_\|] = 0. \tag{B.13}$$

Together, (B.11) and (B.13) imply that $W_tQ_\| - W_t^* \xrightarrow{p} 0$. The conclusion then follows from the fact that $\lim_{t\to 0}W_t^* = W_\infty^*$. This complete the proof of Theorem B.1. $\qquad\square$

## B.4 PROOF OF THEOREM B.4

By Proposition B.6, $\mathbb{E}[W_t] - W_t^*$ has one-sided decay with

$$A_t = \frac{2\eta_t}{N}X_tX_t^\mathsf{T}, \qquad B_t = -\Xi_t^*.$$

By Lemma A.7 and (B.6), $\mathbb{E}[A_t]$ satisfies

$$\log\left\|\prod_{r=s}^{t}\left(\mathrm{Id}-2\eta_r\frac{1}{N}\mathbb{E}[X_rX_r^\mathsf{T}]\right)Q_\|\right\|_2 \leq \frac{1}{2}\log\mathbb{E}\left[\left\|\left(\prod_{r=s}^{t}\left(\mathrm{Id}-2\eta_r\frac{X_rX_r^\mathsf{T}}{N}\right)\right)Q_\|\right\|_2^2\right]$$

$$< \frac{C_1}{2} - \frac{C_2}{2}\int_s^{t+1}r^{-\alpha}dr.$$

Applying Lemma A.4 using this bound and (B.7), we find that

$$\|\mathbb{E}[W_t]Q_\| - W_t^*\|_F = O(t^{\alpha-\beta_1}).$$

Moreover, because $\|\Xi_t^*\|_F = O(t^{-\beta_1})$, we also find that $\|W_t^* - W_\infty^*\|_F = O(t^{-\beta_1+1})$, and hence

$$\|\mathbb{E}[W_t]Q_\| - W_\infty^*\|_F = O(t^{-\beta_1+1}).$$

Further, by Proposition B.6, $\mathbb{E}[(W_t-\mathbb{E}[W_t])^\mathsf{T}(W_t-\mathbb{E}[W_t])]$ has two-sided decay with

$$A_t = \frac{2\eta_t}{N}X_tX_t^\mathsf{T}, \qquad C_t = \frac{4\eta_t^2}{N^2}\left[\mathrm{Id}\circ\mathrm{Var}\left(\mathbb{E}[W_t]X_tX_t^\mathsf{T}-Y_tX_t^\mathsf{T}\right)\right].$$

Applying Lemma A.5 with (B.6) and (B.8), we find that

$$\mathbb{E}\left[\|(W_t - \mathbb{E}[W_t])Q_\|\|_F^2\right] = O(t^{\alpha - \beta_2}).$$

By Chebyshev's inequality, for any $x > 0$ we have

$$\mathbb{P}\left(\|W_t Q_\| - W_\infty^*\|_F \geq O(t^{-\beta_1 + 1}) + x \cdot O(t^{\frac{\alpha - \beta_2}{2}})\right) \leq x^{-2}.$$

For any $\epsilon > 0$, choosing $x = t^\delta$ for small $0 < \delta < \epsilon$ we find as desired that

$$t^{\min\{\beta_1 - 1, \frac{\beta_2 - \alpha}{2}\} - \epsilon}\|W_t Q_\| - W_\infty^*\|_F \xrightarrow{p} 0,$$

thus completing the proof of Theorem B.4. $\qquad\square$

## C  ANALYSIS OF NOISING AUGMENTATIONS

In this section, we give a full analysis of the noising augmentations presented in Section 4. Let us briefly recall the notation. As before, we consider overparameterized linear regression with loss

$$\mathcal{L}(W; \mathcal{D}) = \frac{1}{N}\|WX - Y\|_F^2,$$

where the dataset $\mathcal{D}$ of size $N$ consists of data matrices $X, Y$ that each have $N$ columns $x_i \in \mathbb{R}^n, y_i \in \mathbb{R}^p$ with $n > N$. We optimize $\mathcal{L}(W; \mathcal{D})$ by augmented gradient descent with additive Gaussian noise, which means that at each time $t$ we replace $\mathcal{D} = (X, Y)$ by a random dataset $\mathcal{D}_t = (X_t, Y)$, where the columns $x_{i,t}$ of $X_t$ are

$$x_{i,t} = x_i + \sigma_t G_i, \qquad G_i \sim \mathcal{N}(0, 1) \text{ i.i.d.}$$

We then take a step

$$W_{t+1} = W_t - \eta_t \nabla_W \mathcal{L}(W_t; \mathcal{D}_t)$$

of gradient descent on the resulting randomly augmented loss $\mathcal{L}(W; \mathcal{D}_t)$ with learning rate $\eta_t$. A direct computation shows that the proxy loss

$$\overline{\mathcal{L}}_t = \mathbb{E}\left[\mathcal{L}(W; \mathcal{D}_t)\right] = \mathcal{L}(W; \mathcal{D}) + \sigma_t^2 N \|W\|_F^2,$$

which is strictly convex. Thus, the space

$$V_\| := \text{ column span of } \mathbb{E}[X_t X_t^\mathsf{T}]$$

is simply all of $\mathbb{R}^n$. Moreover, the proxy loss has a unique minimum, which is

$$W_t^* = YX^T(\sigma_t^2 N \operatorname{Id}_{n \times n} + XX^T)^{-1}.$$

### C.1  PROOF OF THEOREM 4.1

We first show convergence. For this, we seek to show that if $\sigma_t^2, \eta_t \to 0$ with $\sigma_t^2$ non-increasing and

$$\sum_{t=0}^\infty \eta_t \sigma_t^2 = \infty \qquad \text{and} \qquad \sum_{t=0}^\infty \eta_t^2 \sigma_t^2 < \infty, \tag{C.1}$$

then, $W_t \xrightarrow{p} W_{\min}$. We will do this by applying Theorem 5.1, so we check that our assumptions imply the hypotheses of these theorems. For Theorem 5.1, we directly compute

$$\mathbb{E}[Y_t X_t^\mathsf{T}] = YX^\mathsf{T} \qquad \text{and} \qquad \mathbb{E}[X_t X_t^\mathsf{T}] = XX^\mathsf{T} + \sigma_t^2 N \cdot \operatorname{Id}_{n \times n}$$

and

$$\mathbb{E}[X_t X_t^\mathsf{T} X_t] = XX^\mathsf{T}X + \sigma_t^2(N + n + 1)X$$

$$\mathbb{E}[X_t X_t^\mathsf{T} X_t X_t^\mathsf{T}] = XX^\mathsf{T}XX^\mathsf{T} + \sigma_t^2\left((2N + n + 2)XX^\mathsf{T} + \operatorname{Tr}(XX^\mathsf{T})\operatorname{Id}_{n \times n}\right) + \sigma_t^4 N(N + n + 1)\operatorname{Id}_{n \times n}.$$

We also find that

$$\|\Xi_t^*\|_F = |\sigma_t^2 - \sigma_{t+1}^2| N \left\| Y X^\mathsf{T} \left( X X^\mathsf{T} + \sigma_t^2 N \cdot \mathrm{Id}_{n \times n} \right)^{-1} \left( X X^\mathsf{T} + \sigma_{t+1}^2 N \cdot \mathrm{Id}_{n \times n} \right)^{-1} \right\|_F$$

$$\leq |\sigma_t^2 - \sigma_{t+1}^2| N \| Y X^\mathsf{T} [(X X^\mathsf{T})^+]^2 \|_F.$$

Thus, because $\sigma_t^2$ is decreasing, we see that the hypothesis (5.5) of Theorem 5.1 indeed holds. Further, we note that

$$\sum_{t=0}^{\infty} \eta_t^2 \mathbb{E} \left[ \| X_t X_t^\mathsf{T} - \mathbb{E}[X_t X_t^\mathsf{T}] \|_F^2 + \| Y_t X_t^\mathsf{T} - \mathbb{E}[Y_t X_t^\mathsf{T}] \|_F^2 \right]$$

$$= \sum_{t=0}^{\infty} \eta_t^2 \sigma_t^2 \left( 2(n+1) \| X \|_F^2 + N \| Y \|_F^2 + \sigma_t^2 N n(n+1) \right) = O \left( \sum_{t=0}^{\infty} \eta_t^2 \sigma_t^2 \right),$$

which by (C.1) implies (B.3). Theorem 5.1 and the fact that $\lim_{t \to \infty} W_t^* = W_{\min}$ therefore yield that $W_t \xrightarrow{p} W_{\min}$.

For the rate of convergence, we aim to show that if $\eta_t = \Theta(t^{-x})$ and $\sigma_t^2 = \Theta(t^{-y})$ with $x, y > 0$, $x + y < 1$, and $2x + y > 1$, then for any $\epsilon > 0$, we have that

$$t^{\min\{\beta, \frac{1}{2}\alpha\} - \epsilon} \| W_t - W_{\min} \|_F \xrightarrow{p} 0.$$

We now check the hypotheses for and apply Theorem B.4. For (B.6), notice that $Y_r = \mathrm{Id} - 2\eta_r \frac{X_r X_r^\mathsf{T}}{N}$ satisfies the hypotheses of Theorem A.6 with $a_r = 1 - 2\eta_r \sigma_r^2$ and $b_r^2 = \frac{\eta_r^2 \sigma_r^2}{a_r^2} \left( 2(n+1) \| X \|_F^2 + \sigma_r^2 N n(n+1) \right)$. Thus, by Theorem A.6 and the fact that $\eta_t = \Theta(t^{-x})$ and $\sigma_t^2 = \Theta(t^{-y})$, we find for some $C_1, C_2 > 0$ that

$$\log \mathbb{E} \left[ \left\| \prod_{r=s}^{t} (\mathrm{Id} - 2\eta_r \frac{X_r X_r^\mathsf{T}}{N}) \right\|_2^2 \right] \leq \sum_{r=s}^{t} b_r^2 + 2 \sum_{r=s}^{t} \log(1 - 2\eta_r \sigma_r^2)$$

$$\leq C_1 - C_2 \int_s^{t+1} r^{-x-y} dr.$$

For (B.7), we find that

$$\|\Xi_t^*\|_F \leq |\sigma_t^2 - \sigma_{t+1}^2| N \| Y X^\mathsf{T} [(X X^\mathsf{T})^+]^2 \|_F = O(t^{-y-1}).$$

Finally, for (B.8), we find that

$$\eta_t^2 \mathrm{Tr} \left[ \mathrm{Id} \circ \mathrm{Var} \left( \mathbb{E}[W_t] X_t X_t^\mathsf{T} - Y_t X_t^\mathsf{T} \right) \right] = O(t^{-2x-y}).$$

Noting finally that $\| W_t^* - W_{\min} \|_F = O(\sigma_t^2) = O(t^{-y})$, we apply Theorem B.4 with $\alpha = x + y$, $\beta_1 = y + 1$, and $\beta_2 = 2x + y$ to obtain the desired estimates. This concludes the proof of Theorem 4.1. $\qquad \square$

# D ANALYSIS OF SGD

This section gives the full analysis of the results for stochastic gradient descent with and without additive synthetic noise presented in Sections 6.1 and 6.2. Let us briefly recall the notation. As before, we consider overparameterized linear regression with loss

$$\mathcal{L}(W; \mathcal{D}) = \frac{1}{N} \| W X - Y \|_F^2,$$

where the dataset $\mathcal{D}$ of size $N$ consists of data matrices $X, Y$ that each have $N$ columns $x_i \in \mathbb{R}^n, y_i \in \mathbb{R}^p$ with $n > N$. We optimize $\mathcal{L}(W; \mathcal{D})$ by augmented SGD either with or without additive Gaussian noise. In the former case, this means that at each time $t$ we replace $\mathcal{D} = (X, Y)$ by a random batch $\mathcal{B}_t = (X_t, Y_t)$ given by a prescribed batch size $B_t = |\mathcal{B}_t|$ in which each datapoint

in $\mathcal{B}_t$ is chosen uniformly with replacement from $\mathcal{D}$, and the resulting data matrices $X_t$ and $Y_t$ are scaled so that $\overline{\mathcal{L}}_t(W) = \mathcal{L}(W; \mathcal{D})$. Concretely, this means that for the normalizing factor $c_t := \sqrt{N/B_t}$ we have

$$X_t = c_t X A_t \qquad \text{and} \qquad Y_t = c_t Y A_t, \tag{D.1}$$

where $A_t \in \mathbb{R}^{N \times B_t}$ has i.i.d. columns $A_{t,i}$ with a single non-zero entry equal to 1 chosen uniformly at random. In this setting the minimum norm optimum for each $t$ are the same and given by

$$W_t^* = W_\infty^* = YX^\mathsf{T}(XX^\mathsf{T})^+,$$

which coincides with the minimum norm optimum for the unaugmented loss.

In the setting of SGD with additive noise at level $\sigma_t$, we take instead

$$X_t = c_t(XA_t + \sigma_t G_t) \qquad \text{and} \qquad Y_t = c_t Y A_t,$$

where $c_t$ and $A_t$ are as before and $G_t \in \mathbb{R}^{n \times B_t}$ has i.i.d. Gaussian entries. In this setting, the proxy loss is

$$\overline{\mathcal{L}}_t(W) := \frac{1}{N}\mathbb{E}\left[\|c_t Y A_t - c_t W X A_t - c_t \sigma_t W G_t\|_F^2\right] = \frac{1}{N}\|Y - WX\|_F^2 + \sigma_t^2\|W\|_F^2,$$

which has ridge minimizer $W_t^* = YX^\mathsf{T}(XX^\mathsf{T} + \sigma_t^2 N \cdot \mathrm{Id}_{n \times n})^{-1}$.

We begin in §D.1 by treating the case of noiseless SGD. We then do the analysis in the presence of noise in §D.2.

## D.1 Proof of Theorem 6.1

In order to apply Theorems B.1 and B.4, we begin by computing the moments of $A_t$ as follows. Recall the notation $\mathrm{diag}(M)$ from Appendix A.1.

**Lemma D.1.** *For any $Z \in \mathbb{R}^{N \times N}$, we have that*

$$\mathbb{E}[A_t A_t^\mathsf{T}] = \frac{B_t}{N}\mathrm{Id}_{N \times N} \qquad \text{and} \qquad \mathbb{E}[A_t A_t^\mathsf{T} Z A_t A_t^\mathsf{T}] = \frac{B_t}{N}\mathrm{diag}(Z) + \frac{B_t(B_t - 1)}{N^2}Z.$$

*Proof.* We have that

$$\mathbb{E}[A_t A_t^\mathsf{T}] = \sum_{i=1}^{B_t}\mathbb{E}[A_{i,t}A_{i,t}^\mathsf{T}] = \frac{B_t}{N}\mathrm{Id}_{N \times N}.$$

Similarly, we find that

$$
\begin{aligned}
\mathbb{E}[A_t A_t^\mathsf{T} Z A_t A_t^\mathsf{T}] &= \sum_{i,j=1}^{B_t}\mathbb{E}[A_{i,t}A_{i,t}^\mathsf{T} Z A_{j,t}A_{j,t}^\mathsf{T}] \\
&= \sum_{i=1}^{B_t}\mathbb{E}[A_{i,t}A_{i,t}^\mathsf{T} Z A_{i,t}A_{i,t}^\mathsf{T}] + 2\sum_{1 \le i < j \le B_t}\mathbb{E}[A_{i,t}A_{i,t}^\mathsf{T} Z A_{j,t}A_{j,t}^\mathsf{T}] \\
&= \frac{B_t}{N}\mathrm{diag}(Z) + \frac{B_t(B_t - 1)}{N^2}Z,
\end{aligned}
$$

which completes the proof. $\square$

Let us first check convergence in mean:

$$\mathbb{E}[W_t]Q_\| \to W_\infty^*.$$

To see this, note that Lemma D.1 implies

$$\mathbb{E}[Y_t X_t^\mathsf{T}] = YX^\mathsf{T} \qquad \mathbb{E}[X_t X_t^\mathsf{T}] = XX^\mathsf{T},$$

which yields that

$$W_t^* = YX^\mathsf{T}[XX^\mathsf{T}]^+ = W_\infty^* \tag{D.2}$$

for all $t$. We now prove convergence. Since all $W_t^*$ are equal to $W_\infty^*$, we find that $\Xi_t^* = 0$. By (B.9) and Lemma D.1 we have

$$\mathbb{E}[W_{t+1}] - W_\infty^* = (\mathbb{E}[W_t] - W_\infty^*)\Big(\mathrm{Id} - \frac{2\eta_t}{N}XX^\mathsf{T}\Big),$$

which implies since $\frac{2\eta_t}{N} < \lambda_{\max}(XX^\mathsf{T})^{-1}$ for large $t$ that for some $C > 0$ we have

$$\|\mathbb{E}[W_t]Q_\| - W_\infty^*\|_F \le \|W_0 Q_\| - W_\infty^*\|_F \prod_{s=0}^{t-1}\Big\|Q_\| - \frac{2\eta_s}{N}XX^\mathsf{T}\Big\|_2$$

$$\le C\|W_0 Q_\| - W_\infty^*\|_F \exp\Big(-\sum_{s=0}^{t-1}\frac{2\eta_s}{N}\lambda_{\min,V_\|}(XX^\mathsf{T})\Big). \quad (\mathrm{D.3})$$

From this we readily conclude using (6.1) the desired convergence in mean $\mathbb{E}[W_t]Q_\| \to W_\infty^*$.

Let us now prove that the variance tends to zero. By Proposition B.6, we find that $Z_t = \mathbb{E}[(W_t - \mathbb{E}[W_t])^\mathsf{T}(W_t - \mathbb{E}[W_t])]$ has two-sided decay of type $(\{A_t\}, \{C_t\})$ with

$$A_t = \frac{2\eta_t}{N}X_t X_t^\mathsf{T}, \qquad C_t = \frac{4\eta_t^2}{N^2}\left[\mathrm{Id}\circ\mathrm{Var}((\mathbb{E}[W_t]X_t - Y_t)X_t^\mathsf{T})\right].$$

To understand the resulting rating of convergence, let us first obtain a bound on $\mathrm{Tr}(C_t)$. To do this, note that for any matrix $A$, we have

$$\mathrm{Tr}\left(\mathrm{Id}\circ\mathrm{Var}[A]\right) = \mathrm{Tr}\left(\mathbb{E}\left[A^\mathsf{T}A\right] - \mathbb{E}\left[A\right]^\mathsf{T}\mathbb{E}\left[A\right]\right).$$

Moreover, using the definition (D.1) of the matrix $A_t$ and writing

$$M_t := \mathbb{E}\left[W_t\right]X - Y,$$

we find

$$\left((\mathbb{E}\left[W_t\right]X_t - Y_t)X_t^\mathsf{T}\right)^\mathsf{T}(\mathbb{E}\left[W_t\right]X_t - Y_t)X_t^\mathsf{T} = XA_t A_t^\mathsf{T}M_t^\mathsf{T}M_t A_t A_t^\mathsf{T}X^\mathsf{T}$$

as well as

$$\mathbb{E}\left[\left((\mathbb{E}[W_t]X_t - Y_t)X_t^\mathsf{T}\right)\right]^\mathsf{T}\mathbb{E}\left[(\mathbb{E}[W_t]X_t - Y_t)X_t^\mathsf{T}\right] = X\mathbb{E}\left[A_t A_t^\mathsf{T}\right]M_t^\mathsf{T}M_t\mathbb{E}\left[A_t A_t^\mathsf{T}\right]X^\mathsf{T}.$$

Hence, using the expression from Lemma D.1 for the moments of $A_t$ and recalling the scaling factor $c_t = (N/B_t)^{1/2}$, we find

$$\mathrm{Tr}(C_t) = \frac{4\eta_t^2}{B_t}\mathrm{Tr}\left(X\left\{\mathrm{diag}\left(M_t^\mathsf{T}M_t\right) - \frac{1}{N}M_t^\mathsf{T}M_t\right\}X^\mathsf{T}\right).$$

Next, writing

$$\Delta_t := \mathbb{E}[W_t] - W_\infty^*$$

and recalling (D.2), we see that

$$M_t = \Delta_t X.$$

Thus, applying the estimates (D.3) about exponential convergence of the mean, we obtain

$$\mathrm{Tr}(C_t) \le \frac{8\eta_t^2}{B_t}\left\|\Delta_t Q_\|\right\|_2^2\left\|XX^\mathsf{T}\right\|_2^2$$

$$\le C\frac{8\eta_t^2}{B_t}\left\|XX^\mathsf{T}\right\|_2^2\|\Delta_0 Q_\|\|_F^2 \exp\Big(-\sum_{s=0}^{t-1}\frac{4\eta_s}{N}\lambda_{\min,V_\|}(XX^\mathsf{T})\Big). \quad (\mathrm{D.4})$$

Notice now that $Y_r = Q_\| - A_r$ satisfies the conditions of Theorem A.6 with $a_r = 1 - 2\eta_r\frac{1}{N}\lambda_{\min,V_\|}(XX^\mathsf{T})$ and $b_r^2 = \frac{4\eta_r^2}{B_r a_r^2 N}\mathrm{Tr}\left(X\,\mathrm{diag}(X^\mathsf{T}X)X - \frac{1}{N}XX^\mathsf{T}XX^\mathsf{T}\right)$. By Theorem A.6 we then obtain for any $t > s > 0$ that

$$\mathbb{E}\left[\left\|\prod_{r=s+1}^{t}(Q_\| - A_r)\right\|_2^2\right] \le e^{\sum_{r=s+1}^{t}b_r^2}\prod_{r=s+1}^{t}\left(1 - 2\eta_r\frac{1}{N}\lambda_{\min,V_\|}(XX^\mathsf{T})\right)^2. \quad (\mathrm{D.5})$$

By two-sided decay of $Z_t$, we find by (D.4), (D.5), and (A.9) that

$$\mathbb{E}[\|W_t Q_\| - \mathbb{E}[W_t]Q_\| \|_F^2] = \mathrm{Tr}(Q_\| Z_t Q_\|)$$

$$\leq e^{-\frac{4}{N}\lambda_{\min,V_\|}(XX^\mathsf{T})\sum_{s=0}^{t-1}\eta_s} \frac{\|XX^\mathsf{T}\|_2^2}{N^2}\|\Delta_0 Q_\|\|_F^2 C \sum_{s=0}^{t-1}\frac{8\eta_s^2}{B_s/N}e^{\frac{4\eta_s}{N}\lambda_{\min,V_\|}(XX^\mathsf{T})+\sum_{r=s+1}^{t}b_r^2}. \quad \text{(D.6)}$$

Since $\eta_s \to 0$, we find that $\eta_s \frac{N}{B_s}e^{\frac{4\eta_s}{N}\lambda_{\min,V_\|}(XX^\mathsf{T})}$ is uniformly bounded and that $b_r^2 \leq \frac{4}{N}\lambda_{\min,V_\|}(XX^\mathsf{T})\eta_r$ for sufficiently large $r$. We therefore find that for some $C' > 0$,

$$\mathbb{E}[\|W_t Q_\| - \mathbb{E}[W_t]Q_\| \|_F^2] \leq C' \sum_{s=0}^{t-1}\eta_s e^{-\frac{4}{N}\lambda_{\min,V_\|}(XX^\mathsf{T})\sum_{r=0}^{s}\eta_r},$$

hence $\lim_{t\to\infty}\mathbb{E}[\|W_t Q_\| - \mathbb{E}[W_t]Q_\| \|_F^2] = 0$ by Lemma A.8. Combined with the fact that $\mathbb{E}[W_t]Q_\| \to W_\infty^*$, this implies that $W_t Q_\| \xrightarrow{P} W_\infty^*$.

To obtain a rate of convergence, observe that by (D.3) and the fact that $\eta_t = \Theta(t^{-x})$, for some $C_1, C_2 > 0$ we have

$$\|\mathbb{E}[W_t]Q_\| - W_\infty^*\|_F \leq C_1 \exp\left(-C_2 t^{1-x}\right). \quad \text{(D.7)}$$

Similarly, by (D.6) and the fact that $\frac{\eta_s}{B_s/N} < \infty$ uniformly, for some $C_3, C_4, C_5 > 0$ we have

$$\mathbb{E}[\|W_t Q_\| - \mathbb{E}[W_t]Q_\| \|_F^2] \leq C_3 \exp\left(-C_4 t^{1-x}\right)t^{1-x}$$

We conclude by Chebyshev's inequality that for any $a > 0$ we have

$$\mathbb{P}\left(\|W_t Q_\| - W_\infty^*\|_F \geq C_1 \exp\left(-C_2 t^{1-x}\right) + a \cdot \sqrt{C_3}t^{\frac{1}{2}-\frac{x}{2}}e^{-C_4 t^{1-x}/2}\right) \leq a^{-2}.$$

Taking $a = t$, we conclude as desired that for some $C > 0$, we have

$$e^{Ct^{1-x}}\|W_t Q_\| - W_\infty^*\|_F \xrightarrow{P} 0.$$

This completes the proof of Theorem 6.1. $\qquad\square$

## D.2 PROOF OF THEOREM 6.2

We now complete our analysis of SGD with Gaussian noise. We will directly check that the optimization trajectory $W_t$ converges at large $t$ to the minimal norm interpolant $W_\infty^*$ with the rates claimed in Theorem 6.2. We will deduce this from Theorem B.4. To check the hypotheses of this theorem, we will need expressions for its moments, which we record in the following lemma.

**Lemma D.2.** *We have*

$$\mathbb{E}[Y_t X_t^\mathsf{T}] = YX^\mathsf{T} \qquad and \qquad \mathbb{E}[X_t X_t^\mathsf{T}] = XX^\mathsf{T} + \sigma_t^2 N \,\mathrm{Id}_{n\times n}. \quad \text{(D.8)}$$

*Moreover,*

$$\mathbb{E}[Y_t X_t^\mathsf{T} X_t Y_t^\mathsf{T}] = c_t^4 \mathbb{E}[YA_t A_t^\mathsf{T} X^\mathsf{T} X A_t A_t^\mathsf{T} Y^\mathsf{T} + \sigma_t^2 YA_t G_t^\mathsf{T} G_t A_t^\mathsf{T} Y^\mathsf{T}]$$

$$= \frac{N}{B_t}Y\,\mathrm{diag}(X^\mathsf{T}X)Y^\mathsf{T} + \frac{B_t-1}{B_t}YX^\mathsf{T}XY^\mathsf{T} + \sigma_t^2 NYY^\mathsf{T}$$

$$\mathbb{E}[Y_t X_t^\mathsf{T} X_t X_t^\mathsf{T}] = c_t^4 \mathbb{E}[YA_t A_t^\mathsf{T} X^\mathsf{T} X A_t A_t^\mathsf{T} X^\mathsf{T} + \sigma_t^2 YA_t G_t^\mathsf{T} G_t A_t^\mathsf{T} X^\mathsf{T}$$

$$\qquad + \sigma_t^2 YA_t G_t^\mathsf{T} X A_t G_t^\mathsf{T} + \sigma_t^2 YA_t A_t^\mathsf{T} X^\mathsf{T} G_t G_t^\mathsf{T}]$$

$$= \frac{N}{B_t}Y\,\mathrm{diag}(X^\mathsf{T}X)X^\mathsf{T} + \frac{B_t-1}{B_t}YX^\mathsf{T}XX^\mathsf{T} + \sigma_t^2(N+\frac{n+1}{B_t/N})YX^\mathsf{T}$$

$$\mathbb{E}[X_t X_t^\mathsf{T} X_t X_t^\mathsf{T}] = c_t^4 \mathbb{E}[XA_t A_t^\mathsf{T} X^\mathsf{T} X A_t A_t^\mathsf{T} X^\mathsf{T} + \sigma_t^2 G_t G_t^\mathsf{T} X A_t A_t^\mathsf{T} X^\mathsf{T} + \sigma_t^2 XA_t G_t^\mathsf{T} G_t A_t^\mathsf{T} X^\mathsf{T}$$

$$\qquad + \sigma_t^2 XA_t A_t^\mathsf{T} X^\mathsf{T} G_t G_t^\mathsf{T} + \sigma_t^2 G_t A_t^\mathsf{T} X^\mathsf{T} G_t A_t^\mathsf{T} X^\mathsf{T} + \sigma_t^2 XA_t G_t^\mathsf{T} X A_t G_t^\mathsf{T}$$

$$\qquad + \sigma_t^2 G_t A_t^\mathsf{T} X^\mathsf{T} X A_t G_t^\mathsf{T} + \sigma_t^4 G_t G_t^\mathsf{T} G_t G_t^\mathsf{T}]$$

$$= \frac{N}{B_t}X\,\mathrm{diag}(X^\mathsf{T}X)X^\mathsf{T} + \frac{B_t-1}{B_t}XX^\mathsf{T}XX^\mathsf{T} + \sigma_t^2(2N+\frac{n+2}{B_t/N})XX^\mathsf{T}$$

$$\qquad + \sigma_t^2\frac{N}{B_t}\mathrm{Tr}(XX^\mathsf{T})\,\mathrm{Id}_{n\times n} + \sigma_t^4 N(N+\frac{n+1}{B_t/N})\,\mathrm{Id}_{n\times n}.$$

*Proof.* All these formulas are obtained by direct, if slightly tedious, computation. $\qquad\square$

With these expressions in hand, we can readily check the of conditions Theorem B.4. First, we find using the Sherman-Morrison-Woodbury matrix inversion formula that

$$\|\Xi_t^*\|_F = |\sigma_t^2 N - \sigma_{t+1}^2 N| \left\|YX^\mathsf{T}(XX^\mathsf{T} + \sigma_t^2 N \cdot \mathrm{Id}_{n\times n})^{-1}(XX^\mathsf{T} + \sigma_{t+1}^2 N \cdot \mathrm{Id}_{n\times n})^{-1}\right\|_F \tag{D.9}$$

$$\leq N|\sigma_t^2 - \sigma_{t+1}^2| \left\|YX^\mathsf{T}[(XX^\mathsf{T})^+]^2\right\|_F.$$

Hence, assuming that $\sigma_t^2 = \Theta(t^{-y})$, we see that condition (B.7) of Theorem B.4 holds with

$$\beta_1 = -y - 1.$$

Next, let us verify that the condition (B.6) holds for an appropriate $\alpha$. For this, we need to bound

$$\log \mathbb{E}\left\|\prod_{r=s}^{t}\left(\mathrm{Id} - \frac{2\eta_r}{N}X_r X_r^\mathsf{T}\right)\right\|_2^2,$$

which we will do using Theorem A.6. In order to apply this result, we find by direct inspection of the formula

$$\mathbb{E}[X_r X_r^\mathsf{T}] = XX^\mathsf{T} + \sigma_r^2 N \,\mathrm{Id}_{n\times n}$$

that

$$\left\|\mathbb{E}\left[\mathrm{Id} - \frac{2\eta_r}{N}X_r X_r^\mathsf{T}\right]\right\|_2 = 1 - 2\eta_r \sigma_r^2 := a_r.$$

Moreover, we have

$$\mathbb{E}\left[\left\|\mathrm{Id} - \frac{2\eta_r}{N}X_r X_r^\mathsf{T} - \mathbb{E}\left[\mathrm{Id} - \frac{2\eta_r}{N}X_r X_r^\mathsf{T}\right]\right\|_2^2\right] = \frac{4\eta_r^2}{N^2}\mathbb{E}\left[\left\|X_r X_r^\mathsf{T} - \mathbb{E}\left[X_r X_r^\mathsf{T}\right]\right\|_2^2\right].$$

Using the exact expressions for the resulting moments from Lemma D.2, we find

$$\frac{4\eta_r^2}{N^2}\mathbb{E}\left[\left\|X_r X_r^\mathsf{T} - \mathbb{E}\left[X_r X_r^\mathsf{T}\right]\right\|_2^2\right]$$

$$= \frac{4\eta_r^2}{N^2}\left[\frac{1}{B_t}\mathrm{Tr}\left(X(N\,\mathrm{diag}(X^\mathsf{T}X) - X^\mathsf{T}X)X^\mathsf{T}\right) + 2\sigma_t^2\frac{n+1}{B_t/N}\mathrm{Tr}(XX^\mathsf{T}) + \sigma_t^4\frac{Nn(n+1)}{B_t/N}\right]$$

$$\leq C\eta_r^2.$$

Thus, applying Theorem A.6, we find that

$$\log \mathbb{E}\left\|\prod_{r=s}^{t}\left(\mathrm{Id} - \frac{2\eta_r}{N}X_r X_r^\mathsf{T}\right)\right\|_2^2 \leq \sum_{r=s}^{t}C\eta_r^2 \log\left(\prod_{r=s}^{t}\left(1 - 2\eta_r\sigma_r^2\right)\right) \leq \sum_{r=s}^{t}C\eta_r^2 - 2\eta_r\sigma_r^2.$$

Recall that, in the notation of Theorem 6.2, we have

$$\eta_r = \Theta(r^{-x}), \qquad \sigma_r^2 = \Theta(r^{-y}).$$

Hence, since under out hypotheses we have $x < 2y$, we conclude that condition (B.6) holds with $\alpha = x + y$. Moreover, exactly as in Proposition B.6, we have

$$\Delta'_{t+1} = \Delta'_t\left(\mathrm{Id} - \frac{2\eta_t}{N}\mathbb{E}\left[X_t X_t^\mathsf{T}\right]\right) + \frac{2}{N}\Xi_t^*, \qquad \Delta'_t := \mathbb{E}\left[W_t - W_t^*\right].$$

Since

$$\|\Xi_t^*\|_F = O(t^{-y-1})$$

and we already saw that

$$\left\|\mathrm{Id} - \frac{2\eta_t}{N}\mathbb{E}\left[X_t X_t^\mathsf{T}\right]\right\|_2 = 1 - 2\eta_t\sigma_t^2,$$

we may use the single sided decay estimates Lemma A.4 to conclude that

$$\|\Delta'_t\|_F = O(t^{x-1}).$$

Finally, it remains to bound

$$\eta_t^2 \operatorname{Tr}\left[\operatorname{Id}\circ\operatorname{Var}(\mathbb{E}[W_t]X_tX_t^\mathsf{T} - Y_tX_t^\mathsf{T})\right].$$

A direct computation using Lemma D.2 shows

$$\mathbb{E}\left[\|Y_tX_t^\mathsf{T} - \mathbb{E}[Y_tX_t^\mathsf{T}]\|_F^2\right] = \frac{1}{B_t}\operatorname{Tr}\left(Y(N\operatorname{diag}(X^\mathsf{T}X) - X^\mathsf{T}X)Y^\mathsf{T}\right) + \sigma_t^2 N\operatorname{Tr}(YY^\mathsf{T}).$$

Hence, again using D.2, we find

$$
\begin{aligned}
&\eta_t^2 \operatorname{Tr}\left[\operatorname{Id}\circ\operatorname{Var}(\mathbb{E}[W_t]X_tX_t^\mathsf{T} - Y_tX_t^\mathsf{T})\right]\\
&= \eta_t^2 \operatorname{Tr}\left(\frac{1}{B_t}\mathbb{E}[W_t]X(N\operatorname{diag}(X^\mathsf{T}X) - X^\mathsf{T}X)X^\mathsf{T}\mathbb{E}[W_t]^\mathsf{T}\right.\\
&\qquad \left. + 2\sigma_t^2\frac{n+1}{B_t/N}\mathbb{E}[W_t]XX^\mathsf{T}\mathbb{E}[W_t]^\mathsf{T} + (\sigma_t^2\frac{N}{B_t}\operatorname{Tr}(XX^\mathsf{T}) + \sigma_t^4 N\frac{n+1}{B_t/N})\mathbb{E}[W_t]\mathbb{E}[W_t]^\mathsf{T}\right)\\
&\quad - 2\eta_t^2 \operatorname{Tr}\left(\frac{1}{B_t}Y(N\operatorname{diag}(X^\mathsf{T}X) - X^\mathsf{T}X)X^\mathsf{T}\mathbb{E}[W_t]^\mathsf{T} + \sigma_t^2\frac{n+1}{B_t/N}YX^\mathsf{T}\mathbb{E}[W_t]^\mathsf{T}\right)\\
&\quad + \eta_t^2 \operatorname{Tr}\left(\frac{1}{B_t}Y(N\operatorname{diag}(X^\mathsf{T}X) - X^\mathsf{T}X)Y^\mathsf{T} + \sigma_t^2 NYY^\mathsf{T}\right).
\end{aligned}
$$

To make sense of this term, note that

$$W_\infty^* X = Y.$$

Hence, we find after some rearrangement that

$$\eta_t^2 \operatorname{Tr}\left[\operatorname{Id}\circ\operatorname{Var}(\mathbb{E}[W_t]X_tX_t^\mathsf{T} - Y_tX_t^\mathsf{T})\right] \le C\eta_t^2(\sigma_t^2 + \|\Delta_t\|_F^2),$$

where we set

$$\Delta_t := \mathbb{E}\left[W_t - W_\infty^*\right].$$

Finally, we have

$$\Delta_t \le \Delta_t' + \|W_t^* - W_\infty^*\|_F = O(t^{x-1}) + \Theta(t^{-y}) = \Theta(t^{-y})$$

since we assumed that $x + y < 1$. Therefore, we obtain

$$\eta_t^2 \operatorname{Tr}\left[\operatorname{Id}\circ\operatorname{Var}(\mathbb{E}[W_t]X_tX_t^\mathsf{T} - Y_tX_t^\mathsf{T})\right] \le C\eta_t^2\sigma_t^2 = \Theta(t^{-2x-y}),$$

showing that condition (B.8) holds with $\beta_2 = 2x + y$. Applying Theorem B.4 completes the proof.
$\square$

