# OpenReview forum: "Data augmentation as stochastic optimization"
_ICLR.cc/2021/Conference — Reject_

### Official Review · AnonReviewer3 · 2020-10-25
**Stochastic optimization, theoretical results, update of Monro-Robbins theorem to data augmentation, additive noise and learning rate.**

**Rating:** 7
**Confidence:** 3

**Review:**

**Summary.** Authors present a novel theoretical framework for assessing the effect of data augmentation (e.g. mini batch SGD), noise addition and the learning rate setup in gradient-based optimization with overparametrized models. Despite the analysis is only performed for linear regression, results extend the well-known Monro-Robbins theorem on rates of convergence. The manuscript is a first step for future analysis of the aforementioned techniques with other type of models and/or loss functions.

**Strengths.** The paper is extremely well-written and even being theoretical, authors did an effort for making it fully understandable to non-familiar readers. I particularly recognize that the line of argumentation is thorough and step-by-step decisions and equations are carefully described with intuitive sentences. i.e. first paragraphs in section 4.

The organization of the paper is good as well, first presenting the type of regression model to be analyzed, the fact of augmenting data, second with the noise-additive models and their implications and finally, connecting all the previous insights with the Moneo-Robbins theorem for a general result that authors fit with mini-batch SGD.

Authors recognize the limitations of their analysis since it is focused in linear models, but to me, this is not an issue for this type of paper. The results, focused on three trending methods as data augmentation, learning rate selection and additive noise are elegant, and as authors mention it only relies on first order moments.

I took a look on the appendix demonstrations and extra information, results seems to be reproducible. References are pretty contemporaneous (last two years).

**Weaknesses, Questions & Recommendations.**
The main weaknesses (to me) in the paper are:
[W1]. Data augmentation is described in a very general way during the first 5-6 papers without much description, and, in the last page, authors focus on the particular case of mini-batch SGD. I understand the reasons for doing this, but preferably, a comment on the application to the mini-batch SGD case would help at the beginning.
[W2]. In a similar way as in [W1], I missed some information about the conditions of $D_t$ wrt $D$ in the beginning of section 3.
[W3]. References are “clustered” in the introduction and related work, but later on, not many of them are connected to the main text. So, connections are difficult to establish with the results and the argumentation. One example is in Section 4, where I missed some cases or references about the synthetic additive noise.

Questions:
[Q1]. Could authors enumerate or describe cases/examples of data augmentation different from mini-batch SGD? References would help. (To make an idea)
[Q2]. In pp.4 authors mention a sufficiently smooth function g. How smooth this function must be? How can we measure this sort of smoothness.
[Q3]. Similarly as before, additive noise is referred with a small \sigma_t. Which is the scale of sigma values considered?
[Q4]. I liked when authors expand the previous results to the case of having the expectation term E[X_t X_t^{\top}]. Just a silly question, in this case, could the expectation operator work as a lower bound? Similarly as in free energy methods or similar? If so, maybe working with the lower bound could help for obtaining similar results for non-linear models.
[Q5]. I do not see why V_{||} in Eq. 5.3 is independent of t as mentioned below Eq. (5.4). The expected value remains stable?

Recommendations:
[Rec1]. A bit more of intuitive descriptions in section 4. for Eq. 4.1. and Eq. 4.2 would help to follow posterior explanations.
[Rec2]. Similarly as Rec1, a bit more of description about Eq. 5.12 and the intrinsic time would help. I lost myself a bit on that part.
[Rec3]. There a few “negative” sentences in the last two pages: “Though SGD is not often considered as a form of …” and “the restriction of the present work to linear models is…”. I think rewriting them in a more positive way would be better. This is an opinion.

**Reasons for score.** I vote for accepting the paper at the conference. Despite the fact that the work is limited to linear models, I think the results are interesting and authors demonstrated a deep comprehension of the main strategies for optimization/fitting of models that are nowadays used everywhere. I would appreciate if authors elaborate a bit more during the rebuttal on the questions pointed out and particularly in the examples of data augmentation.

**Post-Rebuttal Comments.** Thanks for the author's response and their dedication during the rebuttal period, I re-read the updated version of the manuscript and authors did an effort for adding extra content and editing based on the questions and recommendations. In my particular review, they answered and solved the questions that I did. I understand the points addressed by the other reviewers and the theoretical limitations of the method. However, I still have a positive opinion about the paper, and I believe that the aforementioned limitations are well indicated in the paper, something that is valuable. For these reasons, I keep my score.

---

> ### Author Response · Authors · 2020-11-11
> **Responses to review**
>
> Thanks for the detailed comments and questions.  We first give a few more examples of how our work applies outside of mini-batch SGD and then address all other points in detail.
>
> [Q1] Our results Theorem 5.1 and 5.2 apply to any augmentation.  In this paper we have explored their implications for additive noise and (noisy) SGD in detail to illustrate the framework.  Our theorems can be easily specialized to any other augmentation (e.g. random projections, MixUp), and we intend to do so for interesting cases in future work.  We will update the paper to show more explicitly how augmentations such as random projections and MixUp fit into our framework (detailed briefly below).
>
> ## How do the results apply to Mixup?
> To implement Mixup, we can take X_t = X A_t, Y_t = Y A_t, where A_t \in R^{N \times B} has i.i.d. columns containing two non-zero random entries equal to 1 - c and c. The locations of these non-zero entries pick out the datapoints that will be mixed, and the mixing coefficient c is usually random and is drawn from a Beta(\alpha_t, \alpha_t) distribution.
>
> Theorems 5.1 and 5.2 apply to this situation, and when specialized would give conditions on the learning rate \eta_t, and the Beta parameter \alpha_t that would guarantee convergence to the minimizer of the relevant proxy loss. Since Mixup is an important example, we prefer to leave a full analysis to future work. Also, in this setting, the parameters \alpha_t are the analogue of the noise level \sigma_t for Gaussian noise.
>
> ## How do the results apply to random projections (as in Dropout or CutOut)?
> Another class of examples to which our results apply is random projection, such as in Cutout. As a model of random projection, we can take X_t = \Pi_t X, Y_t = Y, where \Pi_t is an orthogonal projection onto a randomly chosen subspace. Denoting by \gamma_t = \Tr(\Pi_t) the dimension of this subspace, we get a proxy loss of the form
>
> L_t(W) = ||Y - \gamma_t W X ||_F^2 + \gamma_t (1 - \gamma_t) \frac{1}{n} \Tr(XX^T) ||W||_F^2.
>
> This is already an insightful computation, we feel, since it shows that random projection both adds an \ell_2 penalty on the weights and applies a Stein-type shrinkage factor to the input data as well. In this setting, Theorem 5.1 shows that if \sum_t \eta_t \gamma_t (1 - \gamma_t) = \infty and \sum_t \eta_t^2 \gamma_t(1 - \gamma_t) < \infty, then W_t converges to W_{min} for any initialization. So again, although random projections are rather different from additive noise, our results still directly apply. In this case the role of augmentation strength is played by 1-\gamma_t.
>
> [Q2] Though we keep this discussion heuristic for now, g only needs to be C^4 so that we can apply 3rd order Taylor expansion with remainder.
>
> [Q3] For a fixed function g, the expansions at the end of Section 4 hold in the limit \sigma_t \to 0.  For a concrete function g at a datapoint x, this would require scaling \sigma_t so that g is well approximated by its 3rd order Taylor expansion about x with radius \sigma_t. Thus, size of \sigma_t is related to the size of the 4th derivative of g in a neighborhood of x.
>
> [Q4] This seems like a cool idea but are not sure that we fully understand this suggestion.  Is the idea to use E[X_tX_t^T] as an approximate lower bound for X_t X_t^T?
>
> [Q5] This is a good point. The statement that V_\parallel is independent of t is an additional assumption we imposed in the main text to make the theorem statement simpler. It holds for key examples such as additive noise, random projections, SGD, Mixup. In general V_\parallel can vary with t, but convergence can only occur on the subspace of directions which “intersect V_\parallel infinitely often”.  We explain in Remark B.2 how to formally define this subspace and how to extend the results to this case. We agree that this point should be explained more carefully and will do so in our forthcoming revision.
>
> [W1] In a revision to the paper to come soon, we will add a list of concrete examples of data augmentation and how they fit into our framework which includes additive noise, random projections, mini-batch SGD, and Mixup.
>
> [W2] Though we often consider D_t as related to D in some way, our approach does not impose _any_ conditions on D_t.  The only place where such conditions will play a role is Theorems 5.1 and 5.2, where they appear in two ways.  First, the augmented dataset D_t should satisfy come consistency conditions for (5.5)-(5.8) to hold. Second, the proxy optima W_t^* depend on D_t, meaning that D_t must have some relation to D in order for their limit W_\infty^* to be a desirable optimum for the original dataset.  We will add some text to clarify this.
>
> [W3] In a revision (to come) to the paper, we will add some references in the main body as objects are introduced.
>
> [Rec1, Rec2] Thank you for suggesting this. We will add more intuitive explanations of the descriptions around (4.12), (4.13), and (5.12).
>
> [Rec3] We will modify to provide a more positive framing.

---

### Official Review · AnonReviewer1 · 2020-10-25
**Some necessary discussions about other possible solutions are missing**

**Rating:** 5
**Confidence:** 4

**Review:**

This paper proposes a framework that re-interprets data augmentation as stochastic optimization for a time-varying sequence of objectives. The paper also provides a theoretical analysis of the simple case of over-parameterized linear regression.

Comments:

1. In section 4, the paper claims that the gradient descent will not converge to W_min because gradient descent “cannot see” the directions in the e in the orthogonal complement of the column span of XX^T. However, it’s easy to choose an initial point W_0 which is in the column span of XX^T. Then the gradient descent may converge to W_min.

2. For over-parameterized linear regression, we can directly compute W_min from the data matrix X and Y. The authors should point out the situation where data augmentation is necessary.

3. Thm 5.1 and Thm 5.2 requires some conditions on W_t, which depends on the algorithm rather than the data.

4. I think the authors should add some experiments to verify the effectiveness of their methods. For example, the experiments can compare the performance of gradient descent and data augmentation with Gaussian noise.

---

> ### Author Response · Authors · 2020-11-11
> **Clarification on the role of over-parameterized linear regression**
>
> Thank you for the review and comments, to which we respond in detail below.
>
> ## Why study over-parameterized linear regression? (Points 1,2)
> The reviewer is absolutely correct that there are many ways to solve overparameterized linear regression, such as initializing gradient descent at W_0 = 0. Our main reason for studying such a simple model is therefore not that finding W_min is challenging but rather because it provides an opportunity to understand the effects of sophisticated augmentation schemes on gradient-based optimization.
>
> Therefore, while there are many other ways to perform linear regression which do not require data augmentation, we choose to study gradient descent with data augmentation in order to provide a point of comparison to more complex models such as non-linear features models, kernels, and neural networks. To make this clear in our forthcoming revision, we will add a more detailed explanation of this role of linear regression and to note that GD can be initialized at 0 to reach W_min.
>
> We hope that the reviewer can reevaluate our paper from this viewpoint not as a paper about methods for over-parameterized linear regression, but as a paper on the general practice of data augmentation.  If this allays the reviewer’s concerns about the novelty and relevance of our work, we would respectfully ask him/her to consider improving the rating of this paper.
>
> ## Why do Theorems 5.1 and 5.2 depend on E[W_t]? (Point 3)
> One of the reviewer’s concerns is that the condition (5.8) in Theorem 5.1 involves E[W_t], which depends on the augmentation scheme.   Both Theorems 5.1 and 5.2 analyze the performance of a specific algorithm (gradient descent with data augmentation), so some kind of dependence on the particular augmentation scheme is unavoidable.  However, we agree that the appearance of E[W_t] may be confusing and will explain it more clearly in the revision.  We give a summary as follows:
>
> 1. Theorem 5.1 holds if either (5.7) or (5.8) are true. The stronger condition (5.7) makes no reference to E[W_t]. Thus, Theorem 5.1 provides conditions for convergence that don’t make any reference to E[W_t].
> 2. Condition (5.8) is weaker than (5.7) and is used to derive sharp results on rates. One way to think abou this is that, given an augmentation scheme, we can first study the average trajectory E[W_t]. This is nothing more than the gradient descent trajectory for the deterministic sequence of proxy losses L_t. This is typically much easier than studying the augmented trajectory W_t because the W_t involves the randomness from each step of augmentation. We can then leverage  information about E[W_t] by plugging it into (5.8) to obtain sufficient conditions for the stochastic convergence of W_t.
>
> Thus, we include both versions in Theorem 5.1 to allow a user of our results to separately analyze E[W_t] to obtain stronger results.  In Theorem 5.2, in contrast, we only state the sharper version with rates.  We will revise the paper accordingly to include more explanations of these points.
>
> ## Experiments on Gaussian noise (Point 4)
> Our results for overparameterized linear regression are proved in a fully rigorous way, meaning that experiments would not add additional validation.  Of course, application to neural networks in practice requires some extension and extrapolation of our results, which we chose to leave to later work.

---

> > ### Comment · AnonReviewer1 · 2020-11-13
> > **Response not convincing**
> >
> > 1. I am not convinced by the authors' response to the study of gradient descent with data augmentation. The authors’ viewpoints should be consistent. If their motivation is to “provide a point of comparison to more complex models such as non-linear features models, kernels, and neural networks”, then their work should at least provide some experimental results on the nonlinear case.
> >
> > 2.  It is ok to only provide analysis for the linear model. But since the authors claim that they study a general framework of data augmentation, it would be more convincing that some experiments on the nonlinear cases are used to show the effectiveness of their methods.
> >
> > 3. I don’t get why ‘some kind of dependence on the particular augmentation scheme is unavoidable’. The updates of the W_t’s depend on the input data, label, and the learning rate, why the dependence on W_t’s is unavoidable? Also, it is weird that a theoretical analysis needs assumptions of the algorithms’ intermediate outputs. A typical analysis should only depend on pure input data and labels.

---

> > > ### Author Response · Authors · 2020-11-16
> > > **The role of theory in the linear setting**
> > >
> > > Thanks for the responses, which we respond to point-by-point below.
> > >
> > > **What is the value of studying the linear case? (Points 1, 2)**
> > > We agree with the reviewer that comparing to experimental results for practical networks would be interesting.  However, we view our contribution as setting the theoretical foundations for future work, which requires analyzing models of increasing complexity.
> > >
> > > Specifically, before our work, to our knowledge there was no case where the effect of general data augmentation could be rigorously understood. Understanding more complex models theoretically first requires understanding the linear case, which our results show already exhibits surprisingly complex behavior (which must also be accounted for in any analysis of a more complex model).
> > >
> > > We believe our approach can generalize to models closer to neural networks for the following reason.  In the infinite width limit, neural networks are described by the Neural Tangent Kernel (NTK, Jacot et. al. 2018), which correspond to infinite dimensional nonlinear feature models.  Applying our results to nonlinear feature models directly applies to this regime.  To be more specific, a fixed nonlinear features model f(X; W) = W * g(X) on a dataset D is equivalent to a linear model on the transformed data set g(D), meaning that the augmented dataset D_t transforms to an augmented version g(D_t) of g(D). From this perspective, Theorems 5.1 and 5.2 give fully rigorous results for arbitrary augmentations on nonlinear features models after this transformation of augmentations. In ongoing work, we are pursuing this approach, which requires translating conditions (5.5-5.7) into conditions on the augmentation in input space, and we sketch the outcome for Gaussian noise in Section 4.
> > >
> > > We hope this can allay the reviewer’s concerns about the relevance of our theoretical work to networks in practice.
> > >
> > > **Why does W_t appear in Theorem 5.1? (Point 3)**
> > > We think there was a minor miscommunication -- the dependence on the augmentation scheme here is through X_t and Y_t, not just W_t.
> > >
> > > With regards to E[W_t], in Theorem 5.1 we have provided two possible groups of conditions to be satisfied, i.e. either (5.5, 5.6, 5.7) or (5.5, 5.6, 5.8).  Conditions (5.5, 5.6, 5.7) do not depend on E[W_t], but only on the input data, labels, and augmented versions thereof.  If desired, the reviewer can ignore the version of Theorem 5.1 which uses conditions (5.5, 5.6, 5.8), which will yield a result depending only on input data and labels as requested.
> > >
> > > In applications, it can be useful to use conditions (5.5, 5.6, 5.8) which require a slightly weaker bound (as (5.7) implies (5.8)), which is why we state both versions.  In all cases that we have considered, specializing our results to a specific case yields results which do not depend on E[W_t], as can be seen from Theorems 4.1, 6.1, 6.2.  We expect similar results for random projections, to give an example.
> > >
> > > We hope this clarifies that the optional appearance of E[W_t] in Theorem 5.1 is for convenience of use in some applications, but that the slightly weaker version of the theorem with (5.7) instead of (5.8) remains a valid result on its own.  If we have misunderstood the reviewer’s concern here, we would be glad to understand it more and respond further.

---

### Official Review · AnonReviewer2 · 2020-10-27
**ICLR 2021 Conference Paper753 AnonReviewer2**

**Rating:** 6
**Confidence:** 3

**Review:**

The paper considers stochastic gradient descent with noisy gradients. In contrast to the standard setting (e.g., gradient Langevin dynamics) where additive Gaussian noise is added to the model gradient, this work focuses on additive perturbations of data instances. As a result of this, the optimization objective changes throughout the training process because the data is no longer static/fixed but assumed to be sampled from some distribution governing the perturbation process (see Eq. 3.3).

Section 4 restricts its consideration to multi-output linear models. A review of prior works shows that stochastic gradient descent with Gaussian perturbations of the inputs is in that setting equivalent (under suitable conditions on the loss) to optimizing regularized objective with the squared Frobenius norm of the weight matrix acting as a regularizer. The problem is then how to characterize the relation between the standard deviation of the Gaussian perturbation measure and learning rate. The main contribution is Theorem 5.1 which provides conditions under which the stochastic gradient descent converges to the min-norm solution. Theorem 4.1 is a special case designed to illustrate the relationship between perturbations and learning rate. The main result is further extended in Theorem 6.1 to mini-batch stochastic gradient descent.

# clarity
I find that the paper is to a certain extent well-written. Still there are parts which are difficult to follow and a revision would be required. For example, I have had problems below Eq. 5.3 when referring to $Q_{||}$. It is not clear what is being projected to $V_{||}$. I have lost track at that point and this then complicates understanding of Theorem 5.1 and pretty much everything that follows. Moreover, checking the proofs was equally confusing.

# originality
The title of this work is miss-leading and not really sure why this is the case. I was expecting to see characterizations of more complex data augmentation schemes that span group theory (e.g., Lie group perturbations). Focusing on additive Gaussian noise might be better suited to stochastic gradient descent with noisy gradients. The noise is different from, for example, Langevin dynamics but there are similarities certainly. Given that I am not an expert on numerous settings and versions of stochastic gradient descent it is difficult to assess the originality. Still, I think that the setting with input perturbations is interesting and worth studying. In addition to this, it might be worthwhile to mention in the abstract (if not in the title) that the focus is on linear models.

# quality
I find the theoretical contribution non-trivial and technical. Still I am not completely convinced in the significance. The first reason for this is in the quite robust nature of Theorem 5.1. I find it difficult to assess how realistic the assumptions are in that theorem (see 5.7 and 5.8). The work attempts to give a more readable result in, for example, Theorem 4.1. The confusing part is what happens to assumptions 5.7 and 5.8. Are they assumed to hold?

I am also not convinced about the related work on data augmentation and stochastic gradient descent with noisy gradients. I am unaware of other works on stochastic gradient descent studying this particular case with input space perturbations. Still some previous works that might be related are:
[1] Vicinal Risk Minimization (NIPS). Chapelle et al.
[2] Variance-based Regularization with Convex Objectives (NIPS). Namkoong & Duchi.

I would also expect a detailed analysis of the contributions relative to:
[3] A Kernel Theory of Modern Data Augmentation (ICML). Dao et al.

---

> ### Author Response · Authors · 2020-11-11
> **Clarification on generality of our results**
>
> Thank you very much for your comments and questions as well as the careful reading of our paper. We respond to your specific points below, but let us first say that we completely agree that we should indicate in the abstract that our main technical results are for linear models. We meant to include this but simply forgot to do so and will make this change in our revision.
>
> As the other comments, we would like to clarify what we feel is an important point: while the model we study is simple, our Theorems 5.1, 5.2 are completely general in the sense that they apply to any augmentation, not just additive noise and SGD. That, we feel, is the main novelty of our work: we can study the effects of sophisticated augmentations in our simple setting.
>
> Our goal in emphasizing the special cases of additive noise and (noisy) SGD was only to show in detail some examples of our general framework. But we feel that ultimately this misled the reviewer to conclude that this was the only set of examples our work covers. We will therefore make clear in the revision (coming soon) that this is not the case by explicitly indicating that our work applies to both Mixup and random projections such as Cutout (see below). We intend to explore the practical consequences of our results for these important augmentations in future work.
>
> We address all the reviewer’s comments in detail below, and we hope overall that we have correctly understood and allayed the reviewer’s concern about the novelty and relevance of our work. If the reviewer agrees we would respectfully ask him/her to consider improving the rating of this paper.
>
> ## How do the results apply to Mixup?
> To implement Mixup, we can take X_t = X A_t, Y_t = Y A_t, where A_t \in R^{N \times B} has i.i.d. columns containing two non-zero random entries equal to 1 - c and c. The locations of these non-zero entries pick out the datapoints that will be mixed, and the mixing coefficient c is usually random and is drawn from a Beta(\alpha_t, \alpha_t) distribution.
>
> Theorems 5.1 and 5.2 apply to this situation, and when specialized would give conditions on the learning rate \eta_t, and the Beta parameter \alpha_t that would guarantee convergence to the minimizer of the relevant proxy loss. Since Mixup is an important example, we prefer to leave a full analysis to future work. Also, in this setting, the parameters \alpha_t are the analogue of the noise level \sigma_t for Gaussian noise.
>
> ## How do the results apply to random projections (as in Dropout or CutOut)?
> Another class of examples to which our results apply is random projection, such as in Cutout. As a model of random projection, we can take X_t = \Pi_t X, Y_t = Y, where \Pi_t is an orthogonal projection onto a randomly chosen subspace. Denoting by \gamma_t = \Tr(\Pi_t) the dimension of this subspace, we get a proxy loss of the form
>
> L_t(W) = ||Y - \gamma_t W X ||_F^2 + \gamma_t (1 - \gamma_t) \frac{1}{n} \Tr(XX^T) ||W||_F^2.
>
> This is already an insightful computation, we feel, since it should that random projection both adds an \ell_2 penalty on the weights and applies a Stein-type shrinkage factor to the input data as well. In this setting, Theorem 5.1 shows that if \sum_t \eta_t \gamma_t (1 - \gamma_t) = \infty and \sum_t \eta_t^2 \gamma_t(1 - \gamma_t) < \infty, then W_t converges to W_{min} for any initialization. So again, although random projections are rather different from additive noise, our results still directly apply. In this case the role of augmentation strength is played 1-\gamma_t.
>
> ## How do the results apply to label-preserving transforms (e.g. Lie group permutations)?
> For a 2-D image viewed as a vector x \in R^n, geometric transforms (with pixel interpolation) or other label-preserving transforms such as color jitter take the form of linear transforms R^n \to R^n.  We may implement such augmentations in our framework by setting
>
> X_t = A_t X, Y_t = Y
>
> for some random transform matrix A_t.  We may then apply Theorems 5.1 and 5.2 to study the optimization properties of these transforms.  However, we do not expect interesting results in this setting because we are focused on optimization rather than generalization, which we leave to future work.
>
> (continued in the next comment)

---

> > ### Author Response · Authors · 2020-11-11
> > **Responses continued**
> >
> > ## Conditions 5.7 and 5.8 in Theorem 5.1 and how they appear in Theorem 4.1
> > One of your concerns is that it is difficult to assess how realistic the assumptions 5.7, 5.8 are in Theorem 5.1. We agree that more discussion here is warranted and will add it in our revision. We will clarify three points.
> > 1. Conditions 5.7 and 5.8 constrain the choice of learning rate and augmentation schedule. For any augmentation scheme, it will be possible to choose parameters so that they are satisfied.
> > 2. Our conditions are not only realistic but close to sharp in the setting of stochastic convex optimization. Indeed, the classical Monro-Robbins conditions \sum_t \eta_t = \infty and \sum_t \eta_t^2 < \infty for stochastic optimization of convex objectives are known to be necessary and sufficient for convergence.
> > 3. Finally, we agree with your point that when extrapolating our results to nonlinear models such as neural networks, it is not a priori obvious how the schedules in our Theorem 5.1 compare to those used in practice. We hope to return to this in future work.
> >
> > We will also explain that conditions (5.7) and (5.8) measure the total variance of the gradient noise caused by augmentation.  For any specific augmentation scheme, Theorem 5.1 therefore provides conditions on the schedule under which convergence to the limiting proxy optimum W_\infty^* occurs, at least in our linear setting.
> >
> > The conditions of Theorem 4.1 are direct specializations of the conditions of Theorem 5.1 to the case of additive Gaussian noise.  Specifically, (5.5) follows from the fact that \sigma_t^2 is decreasing, \sum_t \eta_t \sigma_t^2 = \infty corresponds to (5.6), and \sum_t \eta_t^2 \sigma_t^2 < \infty corresponds to (5.7).  We will point this out explicitly in the revision.
> >
> > ## What is the role of Q_\parallel and V_\parallel?
> > Another concern you brought up was that the discussion around Q_\parallel and V_\parallel was not sufficiently clear, and we will happily be more explicit in the revision. Namely, we will explain that Q_\parallel: R^n \to R^n is the orthogonal projection onto the subspace V_\parallel of R^n, where we recall that input datapoints x lie in R^n.
> >
> > ## Adding related works
> > Thank you for bringing the works [1]-[3] to our attention.  We were unaware of [1], which (in our language) studies proxy losses for augmentation; we will add it to the related work.  We see that [2] is also studying a regularized empirical risk minimization problem, but it seems to use a (adversarial) robust risk which is fixed over the course of training, as opposed to our scheduled proxy losses; We are not exactly sure how to phrase the relation between [2] and our work and are wondering if you had a particular connection in mind?
> >
> > We will also make sure to cite [3] in our revision. We see the relation of our work to [3] as follows:
> > 1. While we handle arbitrary augmentations, the results of [3, Section 3] are stated only for certain augmentations whose action on the dataset has finite orbit; though Gaussian noise is mentioned in Appendix B, we are unclear whether the results stated in the finite setting apply to it or other augmentations without finite orbit.  In addition, our framework is flexible enough to handle augmentations such as Mixup which combine data points, while our understanding is that augmentations in [3] modify each data point independently.
> > 2. Section 3 of [3] studies classification, while we consider linear regression, making the implications not directly comparable.
> > 3. In our language, Section 4 of [3] provides a 2nd order Taylor expansion of the proxy loss for a kernel method with data augmented via x \mapsto T(x) for some distribution T(x). It then gives an intuitive interpretation of each term in this expansion in the kernel setting.  In the special case of a finite-dimensional non-linear features map with Gaussian noise, we give a similar expansion at the end of Section 4 (which also generalizes to arbitrary augmentations in the kernel setting). It seems that there is no discussion in Section 4 of convergence of the augmented optimization trajectory, which is the main subject of our work.
> > 4. Section 5.2 of [3] makes the nice observation that input augmentations can be pushed through a feature map / kernel to augmentations in feature space, and is relevant to our work since we train treat arbitrary augmentations given a fixed set of features.

---

### Official Review · AnonReviewer4 · 2020-10-29
**An analytical study of additive noise data augmentation in SGD for linear regression loss**

**Rating:** 5
**Confidence:** 2

**Review:**

The paper is stuffed with mathematical theorems which makes it almost impossible for me to evaluate the contribution of the work without going through the proofs in the appendix, making this paper more suited for a journal than a conference with 8 pages limit in my opinion. They consider additive Gaussian noise as a data augmentation technique for mini-batch SGD over a simple linear regression with Frobenius loss. The main contribution of the work, as far as I understood, is to derive a range of possible annealing learning rate and additive noise power in Theorem 6.2 that can guarantee convergence of SGD to the global minima of the Frobenius linear regression loss which is convex. But I'm not sure about the derivation; for instance, take x=y=0.4 and e=0.3, then $W_t$ does not necessarily converge to $W_{min}$ in Theorem 6.2, right? Or have I missed something?

Anyway, my biggest problem with the paper is the generality of the proposed framework, which only takes into consideration additive noise and depends strictly on the calculation of gradients as the Frobenius loss illustrates. The author tried to explain a rough idea about how to handle a nonlinear case on Page 4, but this approach can be dramatically cumbersome for the deep neural networks as it needs the calculation of Hessian and gradient with respect to the input x even! Besides, the derivations strictly depend on the additive noise power model $\sigma_t$, but the author claims that their framework is applicable to advanced data augmentation techniques such as Mixup. This is not possible in my opinion. For instance, take a data augmentation technique like Mixup which is not totally data-agnostic such as additive noise. As you know, Mixup is applied to each mini-batch after shuffling in each step of SGD, resulting in a convex combination of the samples of different classes.  In other words, there is no parameter like the noise power $\sigma_t$ that you can control during each step of SGD. There is only $\alpha$ which is not quite flexible either in Mixup as it usually takes a value around 0.2.

Therefore, I strongly doubt the universality of the proposed framework to deal with various networks and data augmentation techniques that are mostly used by practitioners nowadays and do not see the paper as a good fit for a conference like ICLR.

After rebuttal: I thank the author for their response and have raised my rating by one after reading the author's response, but my concern still holds unfortunately regarding the whole approach that the author has taken to analyze the effect of data augmentation. Therefore, I'm still not confident about my rating.

---

> ### Author Response · Authors · 2020-11-11
> **Clarification on the generality of our results.**
>
> We thank the reviewer for the comments and questions. The primary concern seems to be with the generality of the results. Here, we feel that there has been a serious miscommunication, which we seek to clarify in the next section and will of course emphasize in our revision.
>
> ## How general are the results?
> The reviewer is certainly correct that our results are only for linear models with Frobenius loss. However, while the model is simple, our Theorems 5.1, 5.2 are completely general in the sense that they apply to any augmentation, not just additive noise. That, we feel, is the main novelty of our work: we can study the effects of sophisticated augmentations in our simple setting. Our techniques can be applied directly to kernel methods and we believe they can be pushed to give non-trivial insights into data augmentation for neural networks as well. We will add a brief discussion of this point along the lines of what we write in #General Models below.
>
> Our goal in emphasizing the special cases of additive noise and (noisy) SGD was only to show in detail some examples of our general framework. But we feel that ultimately this misled the reviewer to conclude that this was the only set of examples our work covers. We will therefore make clear in the revision that this is not the case by explicitly adding a discussion of how our work applies to both Mixup and random projections such as Cutout. We indicate briefly below what our discussion will look like.
>
> We hope that we have correctly understood the reviewer’s comments and that this allays the reviewer’s concerns about the novelty and relevance of our work. If the reviewer agrees we would respectfully ask him/her to consider improving the rating of this paper.
>
> ## How do the results apply to Mixup?
> To implement Mixup, we can take X_t = X A_t, Y_t = Y A_t, where A_t \in R^{N \times B} has i.i.d. columns containing two non-zero random entries equal to 1 - c and c. The locations of these non-zero entries pick out the datapoints that will be mixed, and the mixing coefficient c is usually random and is drawn from a Beta(\alpha_t, \alpha_t) distribution.
>
> Theorems 5.1 and 5.2 apply to this situation, and when specialized would give conditions on the learning rate \eta_t, and the Beta parameter \alpha_t that would guarantee convergence to the minimizer of the relevant proxy loss. Since Mixup is an important example, we prefer to leave a full analysis to future work. Also, in this setting, the parameters \alpha_t are the analogue of the noise level \sigma_t for Gaussian noise.
>
> ## How do the results apply to random projections?
> Another class of examples to which our results apply is random projection, such as in Cutout. As a model of random projection, we can take X_t = \Pi_t X, Y_t = Y, where \Pi_t is an orthogonal projection onto a randomly chosen subspace. Denoting by \gamma_t = \Tr(\Pi_t) the dimension of this subspace, we get a proxy loss of the form
>
> L_t(W) = ||Y - \gamma_t W X ||_F^2 + \gamma_t (1 - \gamma_t) \frac{1}{n} \Tr(XX^T) ||W||_F^2.
>
> This is already an insightful computation, we feel, since it shows that random projection both adds an \ell_2 penalty on the weights and applies a Stein-type shrinkage factor to the input data as well. In this setting, Theorem 5.1 shows that if \sum_t \eta_t \gamma_t (1 - \gamma_t) = \infty and \sum_t \eta_t^2 \gamma_t(1 - \gamma_t) < \infty, then W_t converges to W_{min} for any initialization. So again, although random projections are rather different from additive noise, our results still directly apply. In this case the role of augmentation strength is played by 1-\gamma_t.
>
> ### What is the relevance of this to neural networks?
> In the infinite width limit, neural networks are described by the Neural Tangent Kernel (NTK, Jacot et. al. 2018), which correspond to infinite dimensional nonlinear feature models.  Applying our results to nonlinear feature models directly applies to this regime.  Specializing to this setting as well as generalizing to finite-width neural networks will be the subject of future work.
>
> Another direction of generalization is that, as we mention at the end of Section 4, a fixed nonlinear features model f(X; W) = W * g(X) on a dataset D is equivalent to a linear model on the transformed data set g(D), meaning that the augmented dataset D_t transforms to an augmented version g(D_t) of g(D). Therefore, Theorems 5.1 and 5.2 apply to nonlinear features models after this transformation of augmentations.  Though this transformation is rather complicated, the corresponding specializations of Theorems 5.1 and 5.2 can be simplified, which we intend to pursue in future work.
>
> Finally, though our derivations currently use the gradient of the Frobenius norm loss, this is not essential, and our results can easily be extended to general strongly convex losses.
>
> (continued in the next comment)

---

> > ### Author Response · Authors · 2020-11-11
> > **Responses continued**
> >
> > ## Presence of Theorems
> > We agree the paper has many theorems with proofs in the appendix.  Our work uses these theorems to shed light on data augmentation, a technique which is pervasive in training neural networks.  We believe this makes our work relevant for the ICLR audience.  A few examples of recent work in ICLR with a similar flavor are:
> >
> > Finite Depth and Width Corrections to the Neural Tangent Kernel, B. Hanin and M. Nica, ICLR 2020.
> > Neural tangent kernels, transportation mappings, and universal approximation, Z. Ji, M. Telgarsky, and R. Xian, ICLR 2020.
> > Beyond Linearization: On Quadratic and Higher-Order Approximation of Wide Neural Networks, Y. Bai and J. Lee, ICLR 2020.
> >
> > ## Convergence rates in Theorem 6.2
> > Theorem 6.2 applies to any choices of x, y, \eps satisfying the stated constraints.  When x = y = 0.4, choosing \eps very small (for example \eps = 0.01) instead of 0.3 shows that |W_t - W_{min}|_F = O(t^{-0.19}).

---

### Author Response · Authors · 2020-11-12
**Revised version of paper**

We have uploaded a revised version of our paper incorporating the comments from all reviewers and making the changes promised in our responses to the reviews.  A particular highlight is the addition of several explicit examples (Gaussian noise, SGD, random projections, label-preserving transforms, Mixup) of how specific augmentations fit into our framework on page 3.

We respectfully ask the reviewers to take a look at the revised paper when considering our responses.

---

### Decision · Program_Chairs · 2021-01-07
**Final Decision**

**Decision:**

Reject

**Comment:**

This paper studies the behavior of SGD for linear models fit with the squared Euclidean loss. There are three main results:

The first result (Sec. 4) studies the behavior where instead of regularizing the objective, Gaussian noise is added to the inputs. The main result is a sufficient condition for how the learning rate and noise can jointly change over time in order for SGD on the MSE error with noisy input to asymptotically converge to the same solution as regular gradient descent without noisy input.

The second result (Sec. 5) is slightly more general in that is considers the case where the noise can be an isotropic Gaussian where the variance changes over time. Again, a result is given for how the learning rate interacts with the data in order to asymptotically converge to the unregularized solution. This is first studied in Thm 5.1 then assuming power-law decay in the noise in Thm. 5.2. It should be emphasized that though these are asymptotic guarantees, the results give asymptotic *rates* of convergence. In my opinion this is a significant strength of the results that was not emphasized by the reviewers.

The third result (Sec. 6) studies SGD for least-squares linear models where the stochasticity is due to data subsampling only. The fraction of subsampled data may change over time.

The primary sentiment from reviewers was that the mathematical complexity of the paper meant that they could not understand it or give a fair review. (More on this below.) For this reason, and because the overall reviews are somewhat borderline, I read the paper in detail. A specific concern raised by two reviewers was that the paper first presents a very general framework but then studies very restricted specific problems. Some reviewers felt that the paper was very well-written, while others felt it was poorly written. There are no experiments.

For my part, I mostly concur that the paper is well-written (albeit quite technical). However, I agree with the concern from reviewers that the technical results all concern extremely restricted settings, and it's not clear what value the extremely general setup brings. I also find the title of the paper a bit puzzling. For specifics of the results, the practical value of Sections 4 and 5 is unclear. It's well-known that adding data noise is exactly equivalent to adding ridge regularization when doing linear regression. But ridge regularized linear regression would be a non-stochastic problem. So what is the value of studying the convergence rates in this case? The paper never makes this clear.

I have concerns about the exponential convergence rate in Thm. 6.1. The paper claims that an exponential convergence rate for SGD has been extensively studied. I do not believe this is true. In general SGD does not have an exponential convergence rate. There are modified methods like SAG that achieve this on finite data sets, but that's not what's studied here. The paper cites two papers: The first is Bottou et al. (2018). This is a lengthy review, with no specific reference given. I am familiar with it and also spent time searching but could not find a specific result. Ma et al. (2018) is also cited. This indeed gives an exponential convergence rate but assuming that at the optimum the loss for all datapoints is zero! No such assumption is made in the submitted paper, and the issue is not further discussed. This is cause for grave concern.

---

> ### Author Response · Authors · 2021-01-14
> **Significant misinterpretations in this reading**
>
> We thank the program chair for reading our paper, but the resulting review contains significant high-level misinterpretations.  We were disappointed that the decision was made on the basis of these confusions, which we would like to record here:
> 1. About the statement "The second result (Sec. 5) is slightly more general in that is considers the case where the noise can be an isotropic Gaussian where the variance changes over time."  This is a substantial mis-reading of our main results (Theorems 5.1 and 5.2). Our results cover arbitrary augmentations, which need not use Gaussian noise or be additive noise augmentations at all.  For example, random projections (e.g. CutOut), SGD, and Mixup as well as their combinations are covered by these results, as we point out when giving these examples at the bottom of page 3.
> 2. About the statement “It's well-known that adding data noise is exactly equivalent to adding ridge regularization when doing linear regression.” This is not the case, as we point out in Section 4 and our citation of Bishop 1995. It is true that, on average over the randomness in the noise, injecting noise is equivalent to ridge regularization. However, a key point of our paper (explained in Section 3 and appearing in the title) is that injecting noise is a _stochastic_ algorithm for this _deterministic_ optimization problem. Moreover, as we explain in detail at the start of Section 4, this achieves more than ridge regression, since simultaneous scaling of noise strength and learning rate allows convergence to the minimal norm solution, which is _not_ achievable by ridge regression with any fixed ridge penalty.  As a result, we want to study convergence rates for this procedure.
> 3. The condition that “at the optimum the loss for all datapoints is zero” in the exponential convergence of SGD from Ma et al., 2018 is automatic in our setting. As we repeatedly note, all our technical results are in the context of overparameterized linear regression in which perfect interpolation is guaranteed at any minimum of the loss, meaning this is not a concern for our Theorem 6.1.

---

> > ### Comment · Area_Chair1 · 2021-01-14
> > **clarity**
> >
> > Regarding overparameterization: I appreciate the clarification and acknowledge that this is mentioned in the paper. However, let me plead for some sympathy for the reader. The abstract of the paper states that overparameterization is only applied as a specialization for a "simple model" without giving clear scope for what that is. The paper clearly states that all analysis is for overparameterized models only in the 5th paragraph on p. 3. It's made easier to miss this because the theorems do not state their assumptions. The citation to Bottou et al. after Thm 6.1 is particularly confusing-- The text states that Bottou provide an exponential convergence rate for SGD, which is (as far as I know) not true.
> >
> > It may be helpful to interpret any misconceptions as suggestions to improve the clarify of the paper. In particular, I suggest to pick a single "scope" where the technical results are given rather than starting with a general framework and specializing it. I *strongly* suggest to make sure all assumptions are included in the Theorem statements, rather than assuming the reader will have collected them from wherever they might occur in the text beforehand.